# Polyvinyl chloride degradation by a bacterium isolated from the gut of insect larvae

Zhe Zhang[1,2,3,4], Haoran Peng[1,2], Dongchen Yang[5], Guoqing Zhang[1,2], Jinlin Zhang[5] & Feng Ju [1,2,3,4,6] ✉

Evidence for microbial degradation of polyvinyl chloride (PVC) has previously been reported, but little is known about the degrading strains and enzymes. Here, we isolate a PVC-degrading bacterium from the gut of insect larvae and shed light on the PVC degradation pathway using a multi-omic approach. We show that the larvae of an insect pest, *Spodoptera frugiperda*, can survive by feeding on PVC film, and this is associated with enrichment of *Enterococcus*, *Klebsiella* and other bacteria in the larva's gut microbiota. A bacterial strain isolated from the larval intestine (*Klebsiella* sp. EMBL-1) is able to depolymerize and utilize PVC as sole energy source. We use genomic, transcriptomic, proteomic, and metabolomic analyses to identify genes and proteins potentially involved in PVC degradation (e.g., catalase-peroxidase, dehalogenases, enolase, aldehyde dehydrogenase and oxygenase), and propose a PVC biodegradation pathway. Furthermore, enzymatic assays using the purified catalase-peroxidase support a role in PVC depolymerization.

Global accumulation of plastic waste and the associated pollution are serious environmental and socioeconomic problems[1]. Polyvinyl chloride (PVC) is a widely used plastic polymer, along with polyethylene (PE), polystyrene (PS), polypropylene (PP), polyurethane (PUR) and polyethylene terephthalate (PET). Among these polymers, the market share of PVC (10.0%) is only lower than those of PE (29.7%) and PP (19.3%) based on European polymer demand[2]. Landfill and incineration processes are commonly used for the treatment and final disposal of plastic waste. However, these technologies are energy-intensive and release secondary pollutants (e.g., chloride and dioxins) and greenhouse gases[3]. Therefore, it is important to develop alternative approaches for a more sustainable and eco-friendly treatment of plastic waste.

Biological degradation and upcycling constitute a promising approach for sustainable treatment of plastic waste and the future development of a relevant green bioeconomy[4,5]. For this purpose,

plastic biodegradation strains and enzymes should be identified, and intermediate biodegradation products should be recovered to be used as alternative chemicals or for biomass production[6]. However, current research progress on PVC biodegradation lags behind that on biodegradation of PE[7–11], PET[12–14], and PS[15,16]. Unlike PET, PVC does not have a hydrolyzable ester bond, making its degradation more challenging. Although there are previous reports on the biodegradation of PVC materials (including plasticizers) by microbial consortia[17,18], little is known about the PVC-degrading microbes or biodegradation pathways involved. Several studies have proposed the possibility of PVC biodegradation by several fungal taxa (i.e., Basidiomycotina, Deuteromycota, Ascomycota) or bacterial taxa (i.e., *Pseudomonas*, *Mycobacterium*, *Bacillus*, and *Acinetobacter*) based solely on the observation of morphological and physicochemical changes (e.g., surface damage and molecular weight loss) associated with plastic degradation[19–22],

[1]Research Center for Industries of the Future (RCIF), School of Engineering, Westlake University, Hangzhou 310024 Zhejiang Province, China. [2]Key Laboratory of Coastal Environment and Resources of Zhejiang Province, School of Engineering, Westlake University, Hangzhou 310024 Zhejiang Province, China. [3]Center of Synthetic Biology and Integrated Bioengineering, School of Engineering, Westlake University, Hangzhou 310024 Zhejiang Province, China. [4]Institute of Advanced Technology, Westlake Institute for Advanced Study, 18 Shilongshan Road, Hangzhou 310024 Zhejiang Province, China. [5]College of Plant Protection, Hebei Agricultural University, Baoding 071000, China. [6]Center for Infectious Disease Research, Westlake Laboratory of Life Sciences and Biomedicine, Hangzhou 310024 Zhejiang Province, China. ✉e-mail: jufeng@westlake.edu.cn

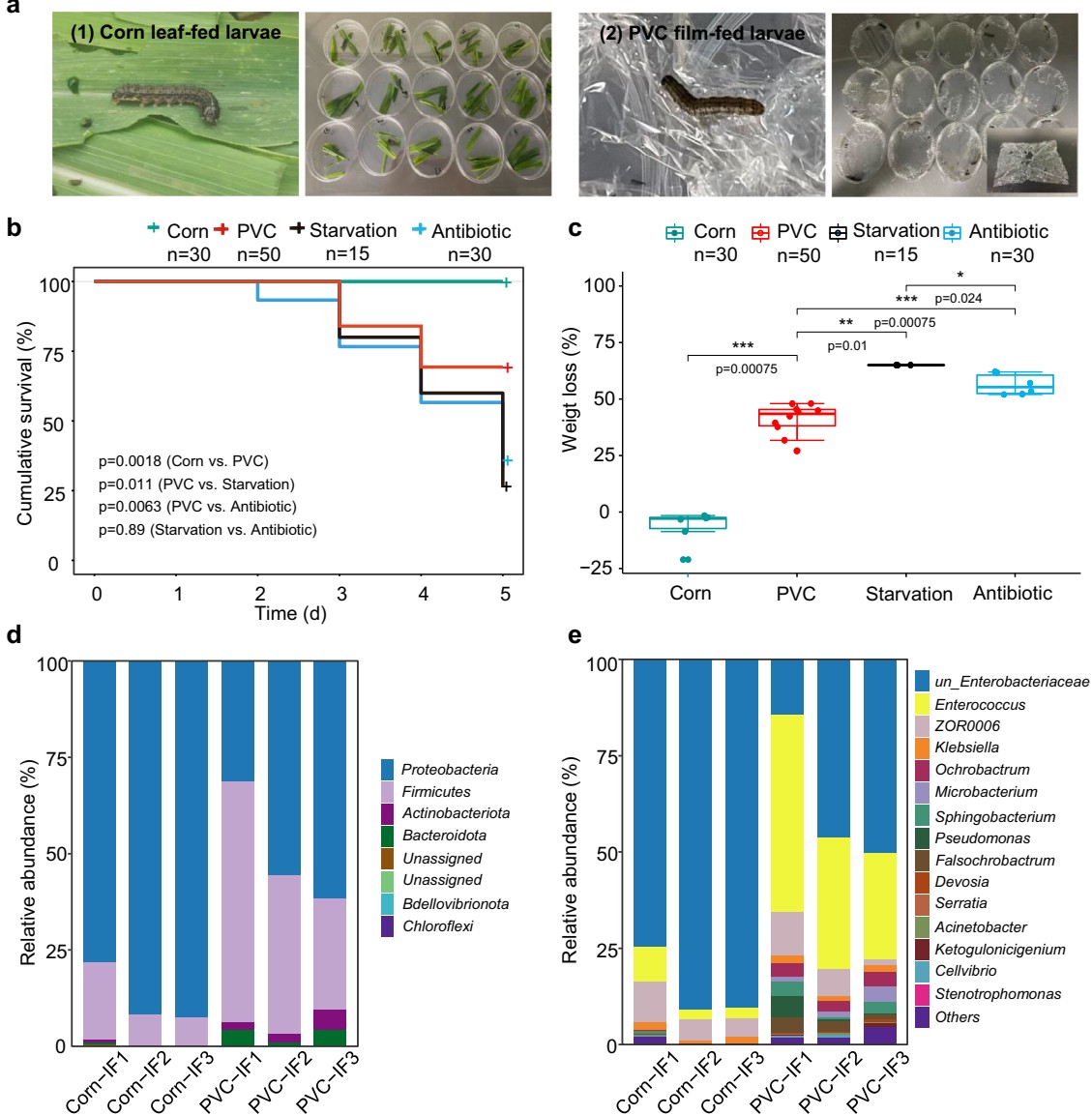

**Fig. 1 | Laboratory cultivation and intestinal microbiota composition of *Spodoptera frugiperda* larvae feeding on corn leaf and PVC film. a** Laboratory setups of replicate cultivation experiments of *S. frugiperda* larva fed with corn leaf (1) and PVC film (2) on petri dishes. **b, c** Cumulative survival and weight loss of larvae in the Corn group, PVC group, Starvation group (no feeding), and Antibiotic group (gentamicin pretreatment of intestinal microbiota before PVC feeding). Log-rank test was used to identify difference in cumulative survival (**b**) between groups. Box plots showed center line as median, and the upper and lower whiskers show maxima and minima, and box limits show upper and lower quartiles. Wilcoxon test was used to identify the difference in weight loss (**c**) between groups. Both statistical tests were two-sided and *P* values (***≤0.001 ≤ **≤0.01 *≤0.05) were adjusted by Benjamini–Hochberg (BH) multiple testing. **d, e** Intestinal microbiota composition at the phylum (**d**) and genus (**e**) levels. Source data are provided as a Source Data file.

and PVC-degrading activity has been reported for an extracellular lignin peroxidase of the fungus *Phanerochaete chrysosporium*[23].

Here, we identify a bacterium that degrades PVC. During laboratory cultivation of the agriculturally invasive insect *Spodoptera frugiperda*, we serendipitously discovered that the larva actively bit and fed on PVC film (Fig. 1a). Motivated by curiosity and recent reports that some insect species (particularly the larvae of wax moths and meal moths[24]) can consume plastic polymers of PE[7,25,26] or PS[16,27], we tested whether *S. frugiperda* larvae could survive solely on PVC film, and whether the larval intestinal microbiota could play a role in film digestion. These studies led to the isolation of a *Klebsiella* strain (named EMBL-1) that can use PVC film as the sole energy and organic carbon source. Furthermore, we explore the PVC biodegradation enzymes, genes, and metabolic products using a multiomic approach, and propose a metabolic pathway for bacterial PVC biodegradation.

## Results

### *S. frugiperda* larva can feed on PVC

To verify our discovery that larvae of *S. frugiperda* can consume PVC film for effective survival, laboratory cultivation experiments were specifically designed and conducted in triplicate to compare the key physiological indexes (i.e., survival rate and body weight) and intestinal microbiota among larvae under starvation (Starvation group, $n = 15$), feeding solely on PVC film (PVC group, $n = 50$), and feeding normally on corn leaves (Corn group, $n = 35$). Overall, the survival rate of the larvae in the PVC group (70%) after 5 days of cultivation was significantly higher than that in the Starvation group (25%) but lower than that in the Corn group (100%, *P* value < 0.0001) (Fig. 1b). This result was consistent with the significant change (*P* value < 0.0001) in the body weight (Fig. 1c) of the larval groups in the following order: starvation (−64.99 (±2.1)%) < PVC (−40.88 ± (10.50)%) < Corn

(6.54(±7.39)%). This finding indicated that PVC film can provide energy for and sustain the survival of the larva, although the growth efficiency of this specialized feeding on PVC film was lower (17%) than that with normal feeding on corn leaf (Fig. 1c). To further test our hypothesis that the intestinal microbiota should be essential for PVC film degradation by *S. frugiperda* larva, we set up the Antibiotic group in which gentamicin antibiotic was used to inactivate most intestinal microbes of the larva. Based on scanning electron microscopy (SEM) analysis (Figure S1), we found the PVC fragments recovered from excreted feces in the PVC group (a) showed strong surface damage in contrast with the Antibiotic group (b), revealing the importance of intestinal microbiota for PVC degradation. These results indicate that *S. frugiperda* larvae can feed on PVC film and suggest a role of the intestinal microbiota in this process.

## Interconnections between the intestinal microbiota and PVC biodegradation

The lack of evidence to date for plastic degradation by germ-free invertebrate larvae generally supports the idea that the intestinal microbiota is the key driver of plastic degradation[7,24]. Supporting our hypothesis on a key role of intestinal microbiota of *S. frugiperda* larva in PVC utilization, we found the survival rate of larva in the Antibiotic group after 5 days was significantly lower (by 33%, Fig. 1b) than PVC group because of gentamicin inhibition of their intestinal microbiota. Accordingly, the body weight change (Fig. 1c) of the PVC-fed larval group treated with gentamicin pretreatment were significantly lower than those of the group without gentamicin pretreatment. The microbial biomass in the larval intestinal microbiota of the Corn group remained stable during the experiment (from $3.90 \pm 0.58$ to $3.70 \pm 0.63 \times 10^6$ CFUs/piece), while the Antibiotic group showed an over 99% reduction in microbial biomass. In addition, SEM analysis of the PVC fragments recovered from excreted feces in the Antibiotic group (Figure S1b) did not show obvious surface changes except for some chew marks at the edges. These results suggest the large dependence of PVC film biodegradation on the larval intestinal microbiota.

PVC film degradation by the intestinal microbiota must lead to the release of transformation products, which we hypothesize should create new ecological niches for microbiome selection through cross-feeding. Supporting our hypothesis, 16 S rRNA gene amplicon sequencing analysis of the larvae intestinal microbiota showed that PVC film degradation triggered a compositional shift from a Proteobacteria-predominated microbiota ($87.5 \pm 8.0\%$ to $49.5 \pm 16.0\%$) to one co-dominated by Firmicutes ($11.9 \pm 7.2\%$ to $44.2 \pm 17.0\%$) (Fig. 1d). Further cross-group comparisons at the levels of bacterial genus and amplicon sequence variant (ASV) showed that compared with normal feeding with corn leaves, PVC feeding and biodegradation not only increased the alpha diversity of ASVs in the larval gut microbiota (i.e., Shannon's H index increased from 0.7 to 2.0, and observed species index increased from 30.0 to 70.0), but also largely favored the selective enrichment of unclassified *Enterococcus* ($4.7 \pm 3.7\%$ to $37.0 \pm 19.8\%$), *Ochrobactrum* ($0.1 \pm 0.2\%$ to $3.4 \pm 0.7\%$), *Falsochrobactrum* (ND to $2.80 \pm 1.40\%$), *Microbacterium* ($0.03 \pm 0.05\%$ to $2.23 \pm 1.55\%$), *Sphingobacterium* ($0.21 \pm 0.30\%$ to $2.55 \pm 1.54\%$) and *Klebsiella* ($1.4 \pm 0.6\%$ to $1.7 \pm 0.4\%$) (Fig. 1e). In summary, PVC film feeding leads to a substantial structural (i.e., diversity and composition) shift in the intestinal bacteriome from phylum down to genus and ASV (a proxy for species) levels.

## Discovery of strain EBML-1 as a PVC-degrading bacterium

Because the larval intestinal microbiota of *S. frugiperda* is associated with PVC film degradation, we assumed that the larval intestine represents an important reservoir of PVC-degrading strains. During laboratory screening, a gram-negative strain (Figure S2a), named *Klebsiella* sp. EMBL-1, formed a visible biofilm on the surface of the PVC

film after only 10 days of incubation (Fig. 2a), causing visible cracks on the surface of the PVC film (Fig. 2b), accompanied by an increase in the strain biomass concentration, i.e., $OD_{600}$ from 0.20 to about 0.60. The cracks formed during initial film degradation might facilitate further plastic degradation[28]. The strain (NCBI accession ID: MZ475068) is most closely related to *Klebsiella variicola* and *Klebsiella pneumoniae*, based on PCR cloning, sequencing, and phylogenetic analyses of the 16 S rRNA gene (Figure S2b).

Using PVC film as a sole energy source, strain EMBL-1 formed a compact biofilm on the film surface after 90 days of incubation. When the biofilm was removed, additional pits and cracks showing film damage were clearly observed (Figure S3a). Based on the contact angle (Figure S3b) and tensile strength tests (Figure S3c) of PVC films, the surface hydrophobicity and tensile strength of the PVC film cultured with strain EMBL-1 changed significantly. These results together suggested that EMBL-1 damaged the physical integrity of the PVC film. During the experiment, the weight loss of the PVC film inoculated with strain EMBL-1 continued to increase over 90 days, and the final average weight loss of the PVC film reached 19.57% (Fig. 2c). The results of advanced polymer chromatography (APC, Waters, China) showed that compared with the control group, the molecular weight measures, i.e., Mn and Mw, of the PVC film in the strain EMBL-1 group decreased by 12.4% and 15.0%, respectively (Fig. 2d), indicating that the long-chain structure of PVC was depolymerized, producing lower-molecular weight fragments. Moreover, thermogravimetric analysis (TGA/DSC 3+ /1600 HT, Mettler-Toledo, Switzerland) showed that the $T_{max}$ and $T_{onset}$ of the PVC film in the EMBL-1 group decreased from 316 °C to 279 °C and from 273 °C to 253 °C, respectively, while these metrics showed limited changes ($T_{max}$: 316 °C to 310 °C and $T_{onset}$: 273 °C to 265 °C) in the control group (Fig. S3e, f), suggesting that the strain EMBL-1 had attacked the PVC polymer chain and reduced the chemical stability of the PVC film.

## PVC biodegradation products released by strain EMBL-1

We then analyzed the transformation products generated by the strain during PVC degradation. Semiquantitative Fourier-transform infrared spectroscopy (FTIR) microspectrometry (ThermoFisher, Nicolet iS50) analysis showed that the infrared spectrum of the surface of the PVC film inoculated with strain EMBL-1 gradually changed compared with the control group over time (Fig. 2e), forming new functional groups such as hydroxyl (3500–3300 cm$^{-1}$) and -C=C- (1550–1650 cm$^{-1}$), indicating the strain can degrade the PVC film via oxidation. We then run NMR experiments to obtain information on the structures of PVC biodegradation products. The $^1$H NMR and $^{13}$C NMR spectrum of pure PVC in EMBL-1 and Control group showed that some substances had been newly produced in the position of 6.5-7.5 ppm ($^1$H spectrum) and 128 ppm ($^{13}$C spectrum), suggesting the products have a functional group of -C=C- (Fig. 2f) and this is consistent with the results of FTIR. The result of Diffusion Ordered Spectroscopy (DOSY) in the NMR experiments showed the same trend as that of APC, demonstrating not only that molecular weight of PVC film in the EMBL-1 group was lower (-12%) than that of the Control group, but also that the molecular weight of the newly produced substance was equivalent to the average molecular weight of the EMBL-1 group (Figure S4a). The result of 2D $^1$H-$^1$H Correlation SpectroscopY (COSY) indicated the H on the -C=C- was a near-correlated relationship and the structure should be -CH=CH- (Figure S4b). The result of 2D $^1$H-$^1$H Nuclear Overhauser Effect SpectroscopY (NOESY) indicated the H on the -CH=CH- functional group had a relationship with the -CH$_2$-CH$_3$- structure (Figure S4c). Moreover, the results of 2D $^1$H-$^{13}$C Heteronuclear Singular Quantum Correlation (HSQC) spectrum and 2D $^1$H-$^{13}$C Heteronuclear Multiple Bond Correlation (HMBC) spectrum enabled us to determine the relationship between C on the -CH=CH- functional group and C on the -CH$_2$-CH$_3$- structure, which suggested that the structure of the substance might be $-[-CH=CH-CH=CH-]_n$-CH$_2$-CH$_3$- (Fig. S4d, e). This together with the

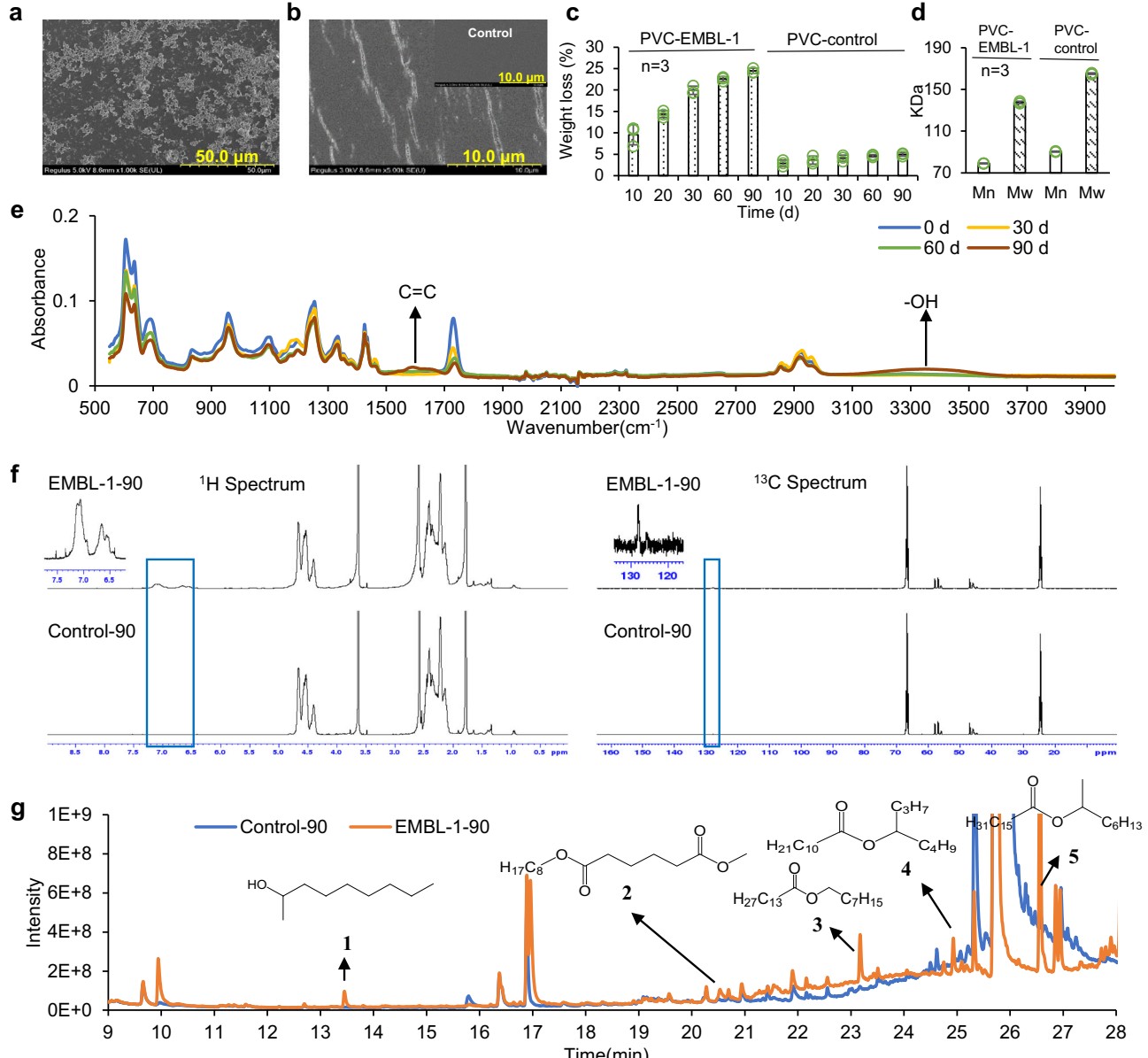

**Fig. 2 | Physicochemical characterization of PVC film degradation by strain EMBL-1. a** The SEM image of the biofilm formed by strain EMBL-1 on PVC film at day 10. **b** SEM image of the cracks formed by strain EMBL-1 on the PVC film at day 10 and the insert image in panel b was the PVC film in the control group. **c, d** The weight loss (**c**) and molecular weight (**d**) of PVC film in PVC-EMBL-1 group (strain EMBL-1 added) and PVC control group (no strain) during 90 days. Each experimental group was conducted in triplicate, the mean values (± standard deviation) were visualized, and source data were provided as a Source Data file. **e** The FTIR results for PVC film in the PVC group (strain EMBL-1 added) over 90 days. **f** The NMR results of PVC film in EMBL-1-90 group (strain EMBL-1 added) and control-90 group (no strain) after 90 days. **g** The detection results of GC-MS of degradation products in PVC films from EMBL-1-90 (strain EMBL-1 added) and control-90 group (no strain) after 90 days.

above results of 2D $^1$H-$^1$H NOESY suggested the structure of the substance as –[-CH=CH-CH=CH-]$_n$-CH(OH)-CH$_2$-, consistent with the generation of hydroxylated products revealed in the FTIR diagram (Fig. 2e). The determination of the structure of the substance also indicated the occurrence of a dechlorination reaction. These data altogether also support PVC degradation by EMBL-1.

We also used GC-MS to quantitatively profile the degradation products of PVC films over 90 days. By comparative inspection of the peaks with significant differences between the EMBL-1 group and the control group, we identified five potential degradation products (compounds 1 to 5) (Fig. 2g), which were sequentially identified as "2-nonanol adipic acid" (1), "adipic acid, methyl octyl ester" (2), "octyl myristate" (3), "dodecanoic acid, isooctyl ester" (4) and "hexadecenoic acid, 1-methylheptyl ester (5)", respectively, according to the high

match score (>800) of each compound in the NIST library (Figure S5). To demonstrate that the above degradation products were not originated from plastic additives, we first identified dioctyl adipate (DOA), dioctyl terephthalate (DOTP), and erucylamide as the three main additives in the PVC film (Figure S6a–e and Table S1). Then, we set up replicated degradation experiments and found that strain EMBL-1 had no ability to effectively degrade the three main additives identified (Figure S6f and Table S1). In addition, the re-cultivation experiments of strain EMBL-1 with soybean oil from the PVC film showed no cell growth (Figure S3d), suggesting that the strain does not derive energy from residual soybean oil (if any) from the pre-cleaned film. Considering the fact that other residual impurities could be detected, we conclude that their influence on PVC degradation by strain EMBL-1 is negligible.

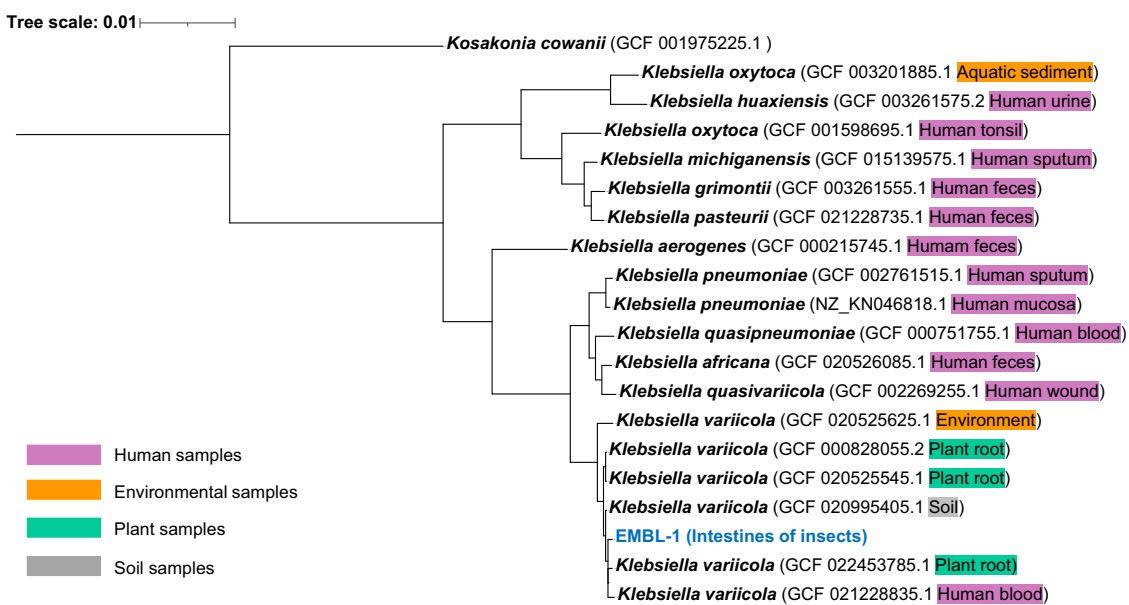

**Fig. 3 | Genome-level phylogeny of strain EMBL-1.** Phylogenetic tree of the strain EMBL-1 was built based on reference genomes from gtdbtk database and visualized with iTOL.

## Genome-level taxonomy and functional profiles of strain EMBL-1

To explore the biodegradation mechanisms of the PVC film, the complete genome of strain EMBL-1 was constructed based on co-assembly of short reads (150 bp × 2) and long reads (average 27,628 bp) derived from Illumina next-generation sequencing and Nanopore third-generation sequencing, respectively. The genome consists of a 5,662,860 bp circular chromosome with a GC content of 57.31%, and 5646 open reading frames (ORFs) predicted to be protein-coding genes. Interactive tree of life (iTOL) analysis with whole-genome strain information of *Klebsiella* extracted from the Genome Taxonomy Database (GTDB) (Fig. 3a) showed that the genome sequence of strain EMBL-1 shared 99.01% average nucleotide identity (ANI) with that of its closest relative, *K. variicola* (RS_GCF_000828055.2).

Functional annotation of strain EMBL-1 genome (Figure S7) was performed by homology-based search against four databases, i.e., NR, GO, KEGG and CAZy. Through KEGG metabolic pathway and network analyses, strain EMBL-1 genome was found to encode 87 genes possibly involved in metabolism and biodegradation of xenobiotics such as phenylacetic acids and 4-hydroxyphenylacetate. Carbohydrate-active enzyme analysis indicated that the genome encodes 74 glycoside hydrolases (GHs), which might contribute to digestion of lignocellulose-containing biomass (e.g., corn leaf) and absorption of carbohydrates by the larvae of *S. frugiperda*. To identify potential PVC-degrading enzymes, we looked for genes encoding enzymes homologous to those known to be involved in degradation of PE[29−36] and PVC[23]. This way we found that strain EMBL-1 had several candidate plastic-degrading genes, annotated as laccase, alkane monooxygenase and peroxidases in the KEGG database (Table 1 and Dataset S1).

## Proteomic analysis of PVC degradation by strain EMBL-1

To obtain further insights into potential enzymes involved in PVC degradation, we carried out a proteomic analysis. Strain EMBL-1 was regrown with PVC film for 30 days before harvesting cells for intracellular (IN) or extracellular (OUT) protein extraction and expression activity tests. The control group supplied with 1% glucose (glu) was set up to better differentiate the metabolic activities for PVC degradation from those for common carbohydrates. The results showed no significant difference in the weight loss of the PVC film between the experimental and control groups (Fig. 4a), revealing that an additional organic carbon source did not improve the PVC degradation efficiency of the strain. Instead, glucose addition only increased the glucose metabolism of the strain EMBL-1 (as evidenced by the following proteomic analysis), which in turn increased the biomass (Figure S8a) and protein levels of the strain (Figure S8b). Moreover, the in vitro activity measurement showed the PVC-degrading activity of the four protein extracts in the following decreasing order: OUT (13.5%) > OUTglu (10.4%) > IN (5.2%) > INglu (5.0%) (Fig. 4b), suggesting that the strain exhibited stronger extracellular than intracellular activities for PVC degradation.

We used LC-MS/MS quantitative proteomics to identify a total of 29 proteins jointly expressed in all four experimental and control groups (Fig. 4c), plus 10 proteins jointly expressed in only the two PVC-degrading experimental groups (Dataset S2). By inspecting the differential expression (DE) of the 39 key proteins between the intracellular (IN) and extracellular (OUT) proteomes, we identified two main categories of proteins associated with PVC degradation. The five most highly expressed extracellular (OUT) proteins (Fig. 4d) included (i) a catalase-peroxidase, an enolase (Eno, with lyase activity), and an aldehyde dehydrogenase (Ad, with redox activity toward aldehyde groups), which could be responsible for the biodegradation of PVC or depolymerized byproducts, and (ii) the highly conserved and universal elongation factor Tu and chaperone protein closely related to microbial translation and protective cell responses to nutrient starvation[37] and heat shock (or other cooccurring harmful materials such as alcohols, inhibitors of energy metabolism, and heavy metal)[38], respectively. Catalase-peroxidases have strong redox capacity and polymer depolymerization ability and have been reported to degrade lignin[39]. Moreover, we identified another five proteins that were more strongly upregulated (defined here as $Log_2(OUT/IN) \geq 3$) extracellularly (OUT) than intracellularly (IN) (Fig. 4d): (i) a dihydroxy-acid dehydratase, which can potentially degrade depolymerized products through cleavage of carbon-oxygen bonds (Fig. 5); (ii) an entericidin EcnA/B family protein, which might be involved in the strain's stress responses to toxic substances (e.g., PVC plasticizers); (iii) porin OmpC and other outer membrane proteins, which might transport some small-molecule metabolites; and iv) glutamate synthase large subunit, which is involved in ammonia assimilation[40].

**Table 1 | Literature review showing a limited understanding of polyvinyl chloride (PVC)-degrading microbial strains and enzymes relative to those for polyethylene (PE)**

| Plastic | Enzyme name | Source | Functional verification | Homologous enzyme in strain EMBL-1 |
|---|---|---|---|---|
| PE | Alkane hydroxylase[29] | *Pseudomonas sp. E4* | Recombinant strains (*E coli*) | -- |
| | Laccase[30,31] | *Rhodococcus ruber* | Crude culture supernatant | Yes |
| | Lignin peroxidase[32] | *Phanerochaete chrysosporium* | Partially purified enzyme used | -- |
| | Alkane monooxygenase[33] | *Pseudomonas aeruginosa* | Recombinant strains (*E coli*) | Yes |
| | Rubredoxin reductase[33] | *Pseudomonas putida* | Recombinant strains (*E coli*) | -- |
| | Manganese peroxidase[34,35] | *Phanerochaete chrysosporium* | Partially purified enzyme used | -- |
| | Soybean peroxidase[36] | Commercial enzyme | Purified enzyme used | |
| PVC | Lignin peroxidase[23] | *Phanerocheate chrysosporium* | Partially purified enzyme used | -- |

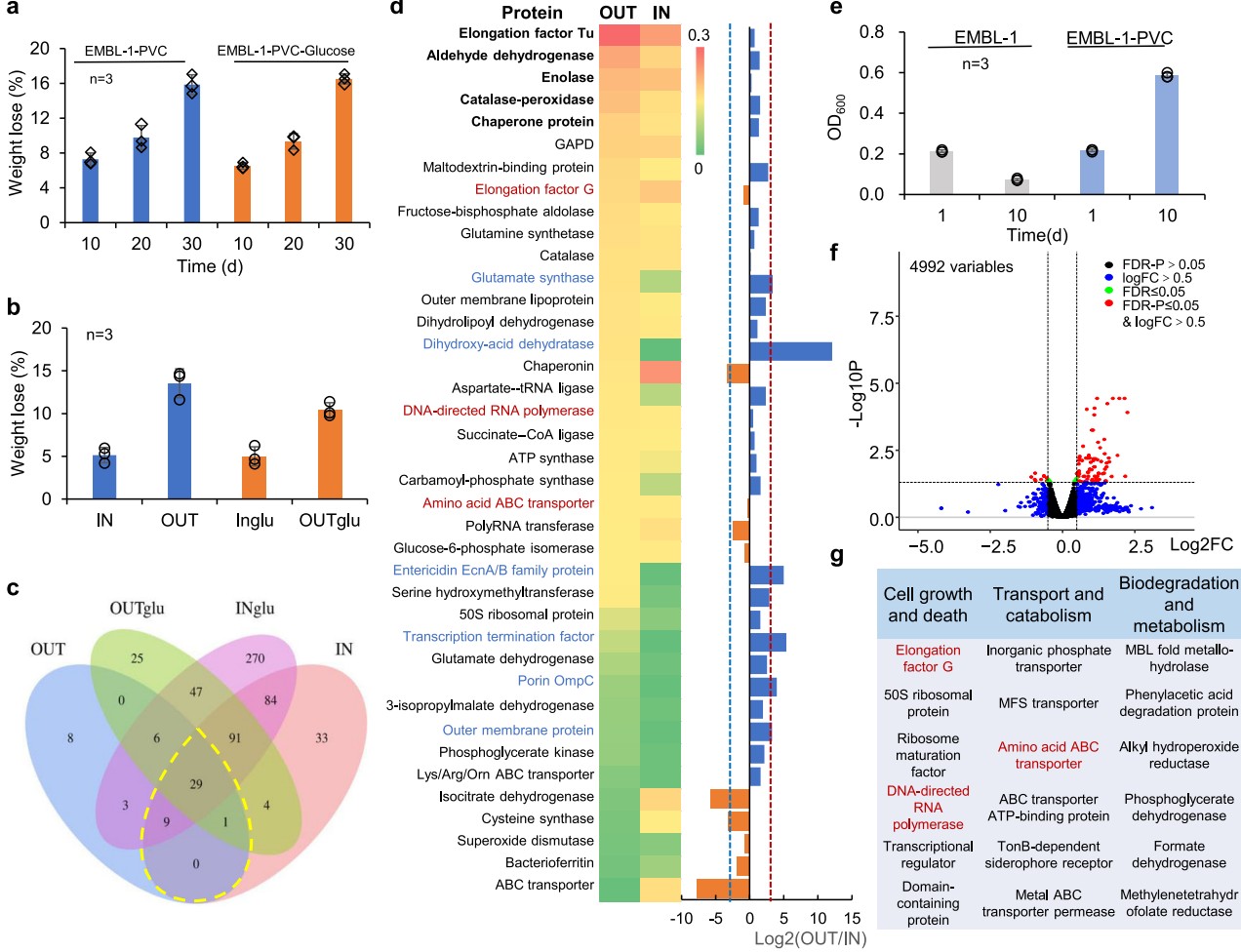

**Fig. 4 | Proteomic and transcriptomic analyses of PVC-degrading metabolic functions of strain EMBL-1. a, b** The weight loss of PVC film under the action of strain EMBL-1 without and with glucose (glu) as supplementary carbon source (**a**) and four intracellular (IN) or extracellular (OUT) protein extracts under in vitro conditions (**b**). **c** Venn diagram showing shared and unique proteins in the protein extracts; **d** 39 proteins involved in intracellular and extracellular metabolism of the PVC film. The heatmap showed the relative abundance of 39 proteins in the proteome, while bar chart depicts just uses the values of fold change from value to visually display the changes in the intracellular and extracellular to intracellular samples. The five most abundant proteins in the OUT extracts from heatmap were marked in bold, while those that were more upregulated in the OUT extracts than in the IN extracts from bar chart depicts (Log2OUT/IN ≥ 8) were marked in blue. The

proteins marked in red were detected in both proteomes and transcriptomes. **e** EMBL-1 cell density (as $OD_{600}$) using PVC film as the sole carbon source. **f** Volcano map showing protein-coding genes (red spheres) that were significantly upregulated in the PVC group compared with the control group. The quasi-likelihood *F* test (glmTreat function in edgeR) was used to identify differential genes in two groups. The Benjamini–Hochberg method was used to adjust *P* values for multiple testing. **g** Functional annotation of upregulated genes, with the genes marked in red being those that were also detected in the proteomes. The proteins marked in red were detected in both proteomes and transcriptomes. For **a, b, e**, each experimental group was conducted in triplicate, the mean values (±standard deviation) were visualized, and source data were provided as a Source Data file.

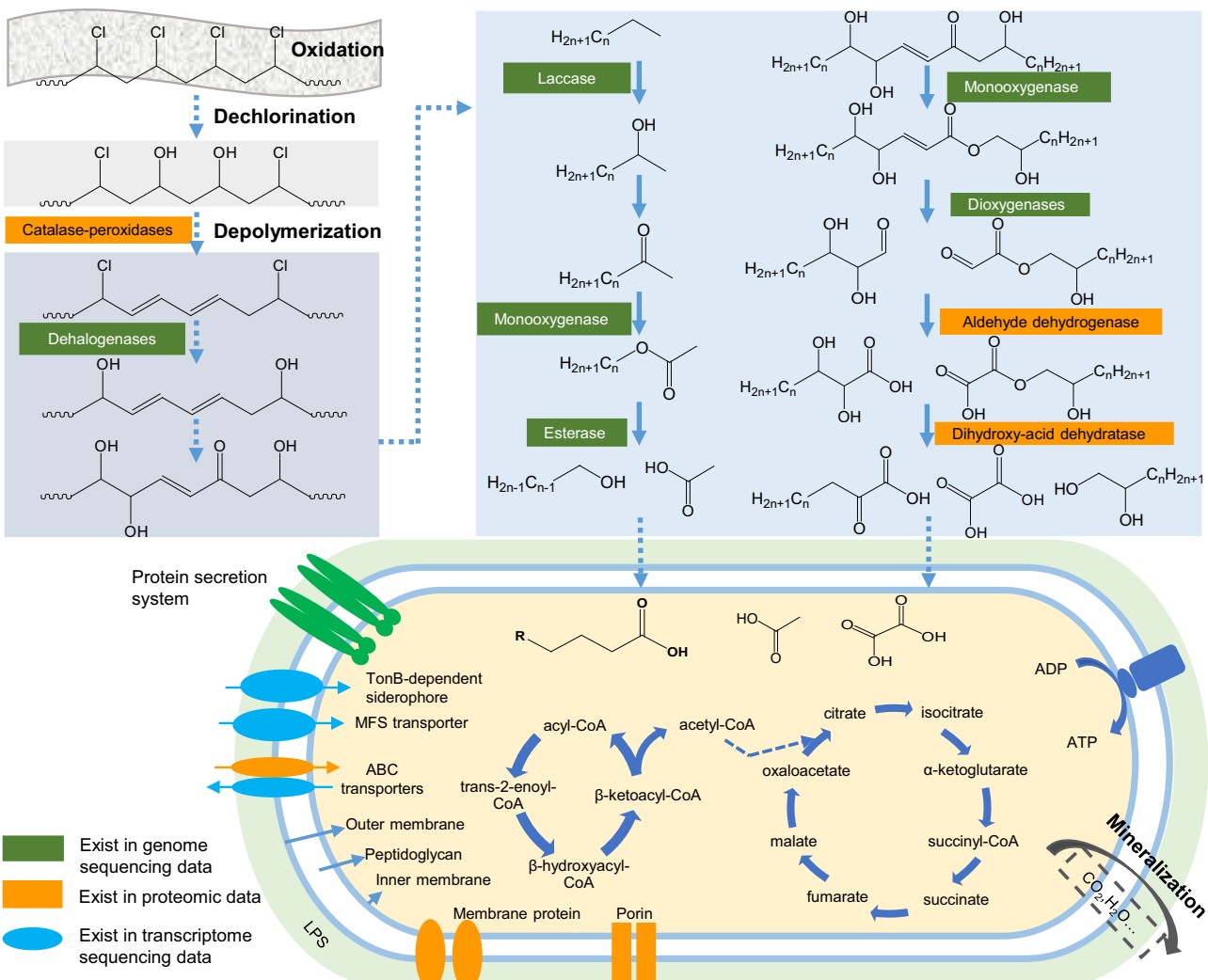

**Fig. 5 | Putative PVC degradation pathway of strain EMBL-1.** The pathway was proposed based on multiomic analyses that integrated the genomic, proteomic, transcriptomic, and metabolomic results for strain EMBL-1 during growth on PVC film. Solid arrows in the figure represented one-step reactions, and dashed arrows represented multi-step reactions. The range of 'n' in the formula of right upper panel was 4-24. The proteins in the green text box and oval were proteins encoded in the genome of EMBL-1. The proteins in the orange text box and oval were the proteins detected in the proteome. The blue oval proteins are the products of genes that were upregulated in the transcriptome. The above proteins are a class of proteins predicted to be involved in the degradation of PVC, and catalase-peroxidase is a functionally verified PVC-degrading enzyme (PVCase) by this study.

## Transcriptomic analysis of PVC degradation by strain EMBL-1

Then we carried out a transcriptomic analysis. Strain EMBL-1 was first grown on PVC film in triplicate for 10 days. The weight loss of the film reached ~7% (Fig. 4a), accounting for an ~threefold increase in the strain biomass (the $OD_{600}$ increased from 0.20 to 0.58, Fig. 4e). We then performed a whole-transcriptomic analysis of strain EMBL-1 to screen 77 out of 96 differentially expressed genes that were significantly (FDR-adjusted $P$ value ≤0.05) upregulated ($\log_2(FC) > 0.5$) or downregulated ($\log_2(FC) < -0.5$) in the PVC group (see red spheres, Fig. 4f) compared with the control group (Dataset S3). Most of the gene expression changes were ascribed to cell growth and death (e.g., elongation factor G, 50 S ribosomal protein, and DNA-directed RNA polymerase), followed by transport and catabolism (e.g., MFS transporter, amino acid ABC transporter, and TonB-dependent siderophore receptor) (Fig. 4g). Among them, three transcriptionally active genes encoded proteins that were also highly represented in the extracellular (OUT) and/or intracellular (IN) proteomes of the strain (see proteins marked in red, Fig. 4d). Notably, genes encoding potential biodegradation-related proteins, such as MBL fold metallohydrolase, phenylacetic acid

degradation protein, and alkyl hydroperoxide reductase, showed upregulated expression during the strain's PVC-dependent growth (Fig. 4g).

## A catalase-peroxidase involved in PVC depolymerization

To study the enzymatic activity of the putative catalase-peroxidase, we expressed the gene in *E. coli* and purified the protein (Figure S8c, d). Enzymatic assays confirmed that the protein displayed catalase (410 U/$mg_{prot}$) and peroxidase (600 U/$mg_{prot}$) activities. We also tested PVC depolymerization activity, and analyzed the PVC in the reaction system. FTIR analysis showed that functional groups -OH and -C=C- had a large vibration, indicating changes in the physiochemical properties of PVC. APC analysis showed that the molecular weight of the PVC in the PVC-catalase-peroxidase group was 12.6% lower than that in the PVC control group (Figure S9a, b), further supporting the depolymerization activity. By comparative inspection of the peaks with significant differences between the PVC-catalase-peroxidase group and the PVC control group (Figure S10a, b), we identified four potential degradation products (Figure S10a, b), identified as "$C_{20}H_{42}$", "$C_{21}H_{44}$", "$C_{24}H_{50}$", and "$C_{25}H_{52}$".

### Proposed pathway for PVC degradation in strain EMBL-1

Based on our genomic, transcriptomic, proteomic, and metabolite analyses, we propose that the following enzymes probably participate in the biodegradation of PVC and its byproducts in strain EMBL-1: catalase-peroxidase, monooxygenase, laccase, lipase, esterase, and carboxylesterase (Table 1 and Dataset S1). This putative pathway would include responses to abiotic factors, extracellular enzymatic depolymerization, and the intracellular metabolism of degradation byproducts (Fig. 5).

Abiotic factors, including light and oxygen, are widely considered to induce plastic degradation reactions that are initiated via C−C and C−H scission[41,42]. We propose that the catalase-peroxidase depolymerizes PVC into lower-molecular weight polymers (Figure S8b). After that, oxidation and enzymatic reactions would further promote degradation of depolymerized PVC products, such as the production of -C=C- bonds and hydroxyl groups (Fig. 5). These factors most likely attack and modify the PVC polymer via hydroxylation, as supported by FTIR. Long-chain products would be transformed into shorter products through a series of enzymatic reactions possibly catalyzed by laccase[31], monooxygenase[43], dioxygenase[44], aldehyde dehydrogenase, esterase, and dihydroxyacid dehydratase (Fig. 5), such as some $C_{24}$-$C_5$ byproducts detected by GC-MS/MS analysis (Fig. 2g). Dioxygenases have been reported to modify and degrade plastic polymers via oxidation of C=C functional groups[44]. Although the expression of laccase, monooxygenase, dioxygenase, esterase, and lipase were below the detection limit of our proteomic analysis, we speculate that these enzymes may have been promptly degraded during PVC-dependent growth of the strain, and/or their expression activities happened to be low at the single sampling points tested in our study. Finally, genes encoding transport and catabolic proteins that were highly represented in the proteome (Fig. 4d) and transcriptome (Fig. 4g) of strain EMBL-1 and might be involved in the transport of small organic molecules and fatty acids to support intracellular catabolism to support the strain's growth.

## Discussion

The limited number of reported PVC-degrading strains originate from environments such as soil[22], landfills[45], and marine environments[18,46], and the underlying biodegradation mechanisms remain unexplored. In the present work, we have isolated a PVC-degrading microorganism from the intestine of insect larvae, and shed light into the enzymes and mechanism potentially involved in PVC degradation. More specifically, we discovered that the larvae of *S. frugiperda* can survive solely on the energy derived from PVC degradation by the intestinal microbiota, and successfully isolated a *Klebsiella* strain (EMBL-1) that degrades PVC. The isolate is most closely related to the species *K. variicola* (ANI > 99%), which is regarded as an emerging pathogen of humans and other animals[47], but also includes plant-associated isolates that can fix nitrogen[48] and degrade xenobiotic pollutants (e.g., atrazine[49]) and natural polymers (e.g., lignin and cellulose[50,51]).

We used a multiomic approach for the identification of PVC biodegradation genes in strain EMBL-1. PVC polymers and monomers are effectively utilized by the strain to derive energy for growth. The strain forms a biofilm on the PVC film (Fig. 2a), which should facilitate its initial destruction of the hydrophobic surface structure of the PVC film by secreted extracellular proteins (e.g., catalase-peroxidase as shown by proteomic analysis, Fig. 4b−d). Our transcriptomic analysis showed upregulation of genes encoding enzymes possibly involved in xenobiotic biodegradation and metabolism (e.g., MBL fold metallohydrolase and alkyl hydroperoxide reductase). We further propose that upregulated transporters (Fig. 4g) may be involved in the uptake of PVC degradation products for intracellular utilization (Fig. 5).

Our findings raise intriguing questions on the evolution of PVC-degrading enzymes and pathways, as previously conceived for PET biodegradation[6]. Plastic-degrading enzymes may have evolved from existing enzymes involved in degradation of natural polymers, such as lignin[39], thus helping microbes and their hosts to use additional nutrients (e.g., plastic feeding by *S. frugiperda* larvae and intestinal strain EMBL-1 as shown here) and adapt to environmental changes (e.g., microplastic pollution). Moreover, we identified at least two groups of genes potentially encoding dehalogenases in the genome of strain EMBL-1, i.e., non-heme chloroperoxidase and HAD family hydrolase genes which are reported to trigger oxidative dechlorination activities of some halogenated compounds[52,53] (Dataset S1). The dehalogenases (i.e. HAD family hydrolase) gene of strain EMBL-1 might be involved in PVC dechlorination. In addition, the genome of strain EBML-1 encodes a monooxygenase and a dioxygenase (Table 1 and Dataset S1) that might participate in oxygen addition to the long carbon chain, as previously shown for similar enzymes involved in PE degradation[54,55], generating free radicals to form carboxyl groups, alcohols, ketones and aldehydes[56]. The oxidation and cleavage of PE make the polymer more hydrophilic, thus facilitating its contact with other extracellular enzymes (e.g., lipase and esterase) after carboxyl group formation or contact of the amide group with endopeptidase[56]. Last but not least, both our proteomic and transcriptomic data show significant expression of MFS and ABC family transporters during PVC degradation by strain EBML-1. Similar transporters play a key role in the uptake of PE degradation products[57].

## Methods

### Field sampling, laboratory cultivation and insect dissection experiments

The larva of *S. frugiperda* were collected from corn fields in Jiangcheng County, Yunnan Province, China. They were then cultured indoors by feeding on corn leaves (in an artificial insect breeding room with a temperature of 25 °C and humidity of 50−60%). To test the hypothesis that the intestinal microbiota of *S. frugiperda* larva should play an essential role in digesting PVC film, which enabled the observed and experimentally verified larval survival on PVC film, 130 4th-instar larva with the same growth status were divided into four groups: (1) the control group (starvation, 15 pcs), (2) the Corn group (fed with corn leaves, 35 pcs), (3) the PVC group (fed with PVC film, 50 pcs), and (4) the Antibiotic group (fed with gentamicin-soaked corn leaves for 3 d and then PVC film, 30 pcs). The body weight of all the numbered larva were measured after 24 h of starvation and after the 5-day experiment. By the end of the experiment, the excreted feces of each experimental group were collected, and the number of survivors was counted. The numbers of culturable cells in the intestinal microbiota in the Antibiotic group and Corn group were also determined.

Subsequently, the surviving larva in the Corn group and PVC group were dissected under aseptic conditions to obtain intestinal samples. Firstly, the survived larva were placed onto ice to weaken their activity. Then the larva were sterilized with 70% alcohol for 2 min and washed three times with sterile water. Afterwards, the head of each larvae was clamped and removed away from the body, and the intestine of the larvae was gently gripped with tweezers and pulled out and placed into a sterile 1.5 ml centrifuge tube. The intestines of every 10 larva in the Corn group or PVC groups were combined and counted as one replicate sample. Three replicates in each group were used for intestinal microbial DNA extraction and sequencing, which were labeled and temporarily stored at 4 °C until further operation. The intestinal feces (IF) samples collected from the PVC and Corn groups were labeled PVC_IF and Corn_IF, respectively. In addition, residual PVC fragments were also recovered from the excreted feces of the PVC group to characterize their surface morphological changes by using a Hitachi field emission scanning electron microscope (Regulus 8230, Japan).

### 16S rRNA gene amplicon sequencing-based analysis of intestinal microbiota

**16S rRNA gene amplicon sequencing.** Total DNA was extracted from intestinal fecal samples collected from the experimental groups using a QIAamp Fast DNA Stool Mini Kit following the manufacturer's recommendations (QIAGEN GmbH, Germany). Then, the hypervariable V4-V5 regions of the prokaryotic 16 S rRNA gene were amplified using 515 F (5′-GTGYCAGCMGCCGCGGTAA-3′) and 926 R (5′-CCGY-CAATTYMTTTRAGTTT-3′). The amplicon products of each sample were evenly mixed and sequenced using a paired-end sequencing strategy (PE250) on the Illumina HiSeq2500.

**Bioinformatic and statistical analyses.** For the 16S rRNA gene amplicon data analysis, FastQC ($v$0.11.9) and cutadapt ($v$1.18) were first used to check the quality of the raw data and excise double-ended primers (fastaq files). Dada2 ($v$1.14) was then used to cluster the input sequence with default parameter settings and for further denoising after importing double-ended data through Quantitative Insight into Microbial Ecology (QIIME2-2020.6) and input-format setting parameters. The next step was to select high-quality (HQ) areas based on FastQC's report results. Taxonomic classification was conducted using the qiime2 built-in package, and the feature-classifier classify-sklearn machine learning method was used for taxonomic annotation using the SILVA 138 SSU as the reference database. The generated files were imported into R studio version 1.1.414 (R version 4.0.3), and phyloseq ($v$1.32.0) was used for statistical analysis and visualization of the data. In addition, the survival ($v$3.2.7) and survminer packages ($v$0.4.9) were used to calculate and draw the survival curve, while ggplot2 ($v$3.3.3) was used to draw box plots of weight. ANOVA was used to check the significance of differences between experimental groups.

### Enrichment, isolation, and identification of PVC-degrading strain EMBL-1

To isolate plastic-degrading microbes, the intestinal materials of ten larva were suspended in 10 mL of PBS and vortexed for 5 min. Then, the intestinal mucosa was removed from the mixed solution. The remaining suspension was used as a microbial inoculum and transferred to a 250-mL flask containing 0.1 g of pre-cleaned PVC film and 100 mL of MSM liquid medium. The PVC film purchased from Sinopec Yanshan petrochemical company (11–12 μm film thickness, about 15% of soybean oil content and some additives of unknown content) was cut into 20 × 20 mm pieces. The PVC pieces were sterilized in 75% ethanol for 20 minutes, washed three times with sterile water and air-dried in an ultra-clean workbench. This cleaning process dissolved and washed way the most soybean oil and impurities in the PVC films. Finally, the cleaned PVC pieces were weighed before the experiment. Then, the cells were cultivated on a shaker (150 rpm/min) at 30 °C and transferred every 15 days. After 45 days, the culture medium was first diluted and then spread onto MSM agar medium plates with PVC film as the sole carbon source for cultivation to enrich PVC-degrading strains. The enriched PVC-degrading strain was further subcultured until a pure colony of the isolate was obtained. Depending on whether the isolates were grown in PVC film-amended liquid MSM, the surface changes of the PVC film were inspected by SEM until a PVC-degrading strain (named EMBL-1) was successfully obtained.

To identify the strain, a near full-length 16 S rRNA gene sequence was PCR amplified using the universal primers 27 F (5′- AGAGTTT-GATCCTGGCTCAG-3′) and 1492 R (5′-GGTTACCTTGTTACGACTT-3′). The 16 S rRNA gene amplicon sequence obtained from Sanger sequencing was deposited in the National Center for Biotechnology Information (NCBI) database and annotated using NCBI's online Basic Local Alignment Search Tool (BLAST) in August 2021.

### Biodegradation of PVC film by strain EMBL-1

Cultivation experiment was conducted to explore the ability of strain EMBL-1 to degrade the cleaned PVC film. Three experimental groups were designed: (1) MSM liquid medium (20 mL) + strain EMBL-1 (OD600 = 0.2); (2) MSM liquid medium (20 mL) + strain EMBL-1 (OD600 = 0.2) + PVC film (weighed); and (3) MSM liquid medium (20 mL) + PVC film (weighed). Each experimental group was prepared in 27 replicates and immediately inoculated with the strain. The following operational procedures were repeated every 10 days: (i) the PVC films from three replicates of each group were removed; (ii) the degradation of the PVC films was examined by SEM, and the films were then weighed; and (iii) half of the MSM liquid medium was replaced. The entire experiment lasted for 90 days, during which period, PVC films were collected and stored on days 10 and 90. The post-treatment of the PVC film was as the following: in order to observe the growth of strain EMBL-1 on PVC film, the PVC film was rinsed properly using sterile water to remove its thick biofilm. Then the cells were collected and sequentially fixed with 2% glutaraldehyde, 25% ethanol, 50% ethanol, 75% ethanol, and 100% ethanol to obtain a good cell fixation. Then some PVC film was mixed with a 2% $w/v$ sodium dodecyl sulfate aqueous solution, shaked for 4 h, and then rinsed with sterile water until the biofilm on the PVC film was completely removed. The cleaned films were weighed after being dried in a drying oven (50 °C) for 24 h. The weight loss (%) was defined as the difference between the weight loss percentage (calculated as 100%×(Initial weight−final weight)/ initial weight) of the EMBL-1 group and the Control group. The downstream morphological and physicochemical characterization methods and precedures were conducted as the following steps.

For the cleaned PVC film, additional degradation experiments were performed to check the degrading activity and capacity of soybean oil. Commercial soybean oil was purchased and used for further activity detection (Jiangsu Aikon Biopharmaceutical R&D Co., Ltd). Then the following experiments were set up in triplicates: (1) MSM liquid medium + EMBL-1, (2) MSM liquid medium+ EMBL-1 + soybean oil (2 mg/L), (3) MSM liquid medium + soybean oil (2 mg/L). All treatments were cultured in a shake (150 rpm, 30 °C) for 70 days, and the growth of EMBL-1 in each group were measured. Additionally, a gas chromatography-mass spectrometry (GC-MS, Trace1300-ISQ7000, ThermoFisher, Singapore) method was established to analyze the composition of additives in the film. The detection conditions are as follows: Pyrolysis/thermal desorption GC-MS (TD/Py-GC-MS) methods were used for quantitation of additives in PVC film using Thermofisher trace1300-ISQ7000. The oven temperature for TD was programmed for 250 °C for 5 min to 350 °C at 20 °C/min, and then held at 1 min at 350 °C. Then the PY temperature was held at 1 min at 610 °C and the interface was set at 300 °C. Sample was injected into the column (TG-5SILMS (30 m, 0.25 mm, 0.25 um) with helium as carrier gas at a constant flow rate of 1 ml min$^{-1}$. The sample was injected at an initial temperature of 50 °C (hold for 2 min) which was progressively increased at 10 °C per minute and held at 320 °C for 3 minutes. Similarly, the detector conditions such as transfer line temperature, ion source temperature, ionization mode electron impact and scan time were maintained at 300 °C, 280 °C, 70 eV, and 0.3 s, respectively. Further, the spectrum attained from the detected compounds at 20–550 Da was compared with GC-MS NIST library of known compounds to identify major additive composition in the PVC film. The test results showed that there were mainly three plasticizers in PVC film, namely DOA, DOTP, and erucylamide. The content of three major additives in the film were measured by GC-MS referring to the national standard method titled "Consumer product-Plastics-Rapid screening of phthalates" (GB/T 39110-2020). The additive chemicals including dioctyl adipate (DOA, purity≥98%, powder), dioctyl terephthalate (DOTP, purity ≥ 99%, liquid), and erucylamide (purity ≥ 99%, liquid) were purchased at Energy Chemical company. Each sample was

injected into the column (same as the above) with helium as carrier gas at a constant flow rate of 1 ml min⁻¹. The sample was injected at an initial temperature of 180 °C (hold for 1 min) which was progressively increased at 10 °C per minute and held at 300 °C (hold for 3 min). Moreover, the detector conditions such as transfer line temperature, ion source temperature, ionization mode electron impact and scan time were maintained at 280 °C, 250 °C, 70 eV, and 0.3 s, respectively. To check whether strain EMBL-1 can degrade the additives in the PVC film, the following experiments were set up in triplicates: (1) MSM liquid medium + EMBL-1 (OD600 = 0.2), (2) MSM liquid medium+ EMBL-1 (OD600 = 0.2) + DOA/DOTP/erucylamide (100 mg/L, 20 mg/L, 5 mg/L), (3) MSM liquid medium + DOA/DOTP/erucylamide (100 mg/L, 20 mg/L, 5 mg/L) All treatments were cultured in a shake (150 rpm, 30 °C) for 30 days, and the growth of strain EMBL-1 in each group was measured every 5 days. The results eliminated the degradation of these substances in the PVC film during the degradation process of the strain.

## Characterization of PVC film damage and biodegradation products

**PVC film damage.** To validate and follow PVC film biodegradation by strain EMBL-1, multiple classic physicochemical methods were used together to analyze the temporal changes in the morphological, compositional, and other physiochemical properties over 90 days. First, colonization by the strain was morphologically characterized by SEM after cell fixation. Moreover, the degradation efficiency of the strain was directly measured based on the weight loss of the PVC film on a 10-day basis. Furthermore, changes in the physical properties of the PVC film were detected by contact angle and tensile strength tests. The contact angle values of PVC film were measured by an automatic contact angle measuring instrument (Dataphysics OCA25, Germany). Changes in tensile strength of PVC film were determined on a universal testing machine (TY8000-A, Tianyuan Testing Machine Co. Ltd., Jiangsu, China) equipped with a 200 N cell. The PVC films (2.5 cm × 0.5 cm) in two groups (collected on 90 d) were tested at room temperature (23 °C) with a relative humidity of (50 ± 2)%️ and 25 mm/min, three replicates in each group. All samples were equilibrated to 50% relative humidity for at least 40 h before analysis. The depolymerization of the plastic materials was also recorded by measuring the change in molecular weight. Advanced Polymer Chromatography (APC) measurements were carried out with three columns XT900-XT450-XT 200 (2.5 μm, 4.6 × 150 mm) and a RI detector. Tetrahydrofuran (THF, HPLC) was used as mobile phase (0.4 ml/min) after calibration with polystyrene standards of known molecular mass. Non-incubated PVC film was used as references. An FTIR microspectrometer (ThermoFisher, Nicolet iS50, China) with a scan range of 4000–500 cm⁻¹ using OMNIC software (v9.12.928) was used to analyze and detect the changes in the surface chemical composition and functional groups of the PVC film (32 scans for each spectrum). Thermogravimetric analysis (TGA/DSC 3+/1600 HT, Mettler-Toledo, Switzerland) was used to compare the initial degradation temperature and the maximum degradation temperature of the PVC film, and the composition, heat stability, and thermal decomposition of the PVC film and the possible intermediate products were examined. Dried PVC films of 5 mg were subjected to thermogravimetric analysis using a Perkin Elmer TGA7 thermal analyzer under a nitrogen atmosphere (gas flow: 40 ml/min). The thermograms were recorded from 50 °C to 800 °C at a heating rate of 10 °C/min. A Bruker NEO 600 MHz NMR spectrometer (600.23 MHz for proton frequency) equipped with a TXI probe and a Bruker NEO 500 MHz NMR spectrometer (500.3 MHz for proton frequency) equipped with a BBO Cryoprobe were used to identify the structure of some degradation products. The residual PVC film (50 mg) in control and EMBL-1 groups were dissolved in tetrahydrofuran (THF) solution (15 mL), and then 75 mL of methanol was added to precipitate the PVC polymer. The precipitates were obtained

by centrifugation and dried. Subsequently, the polymers (precipitates) were resolved in THF-d8 (99.5 atom%, Aladdin–Holdings Group, Beijing) to form solutions of ~20 mg residue/mL and then transferred to NMR tubes for analysis. All NMR experiments were performed at 25 °C on a Bruker NEO 600 MHz NMR spectrometer equipped with a TXI probe and a Bruker NEO 500 MHz NMR spectrometer equipped with a BBO Cryoprobe. For 1D ¹H and ¹³C experiments, 64k complex data points were acquired with 16 and 2880 scans, respectively. 2D ¹H-¹H COSY using the "cosygpppqf" pulse sequence were collected with 2 scans × 2048 data points (F2) × 256 increments (F1). 2D ¹H-¹H NOESY using the "noesygpphpp" pulse sequence were collected with 8 scans × 2048 data points (F2) × 256 increments (F1) with the mixing time of 300 ms. 2D 1H-13C HSQC using the "hsqcedetgpsisp2.3" pulse sequence was collected with 4 scans × 2048 data points (F2) × 256 increments (F1). 2D 1H-13C HMBC using the "hmbcgplpndqf" pulse sequence were collected with 16 scans × 2048 data points (F2) × 256 increments (F1). DOSY experiments were using the "ledbpgp2s" pulse sequence to measure the self-diffusion coefficient D, with a relaxation delay of 3.0 s and 8 scans in total. 16 linear steps from 2% to 95% of gradient strength and a t2 (F2 dimension) of 16k sampling data points were used. The implemented diffusion time big delta and the diffusion gradient length little delta were 80 and 10 ms, respectively. Then, the spectra were analyzed using MestReNova software (version 12.0.0).

**Biodegradation products.** To obtain more evidence for the biodegradation of PVC film by the strain EMBL-1, GC-MS analysis was used to detect potential biodegradation products of PVC in the PVC films. Degraded PVC films weighing 0.03 g were cut into pieces and mixed with 10 mL of THF, and the mixture was ultrasonicated for 30 min at room temperature. The extract was concentrated to 0.5 ml by drying with nitrogen gas and mixed with 1 mL of n-hexane to obtain some possible products by vortex and ultrasonication for 10 min. The samples were filtered using a 0.22 μm PTFE syringe filter for subsequent steps[58]. The sample was injected at an initial temperature of 40 °C (hold, 4 min), which was progressively increased at 10 °C per minute and held at 280 °C (hold, 5 min). Moreover, the detector conditions, i.e., the transfer line temperature, ion source temperature, ionization mode electron impact and scan time, were maintained at 250 °C, 280 °C, 70 eV, and 0.3 s, respectively.

## Whole-genome sequencing analysis of the PVC-degrading strain EMBL-1

To further explore the mechanism underlying the biodegradation of PVC film by strain EMBL-1 and discover the PVC-degrading genes or enzymes involved in the process, the TIANamp Bacteria DNA Kit was used to extract the genomic DNA of the strain, and the genomic DNA of the strain EMBL-1 was split into two fetches and sequenced using both Illumina next-generation sequencing (PE150) and Oxford Nanopore (PromethION). The experimental procedures, including sample quality testing, library construction, library quality testing, and library sequencing, were performed in accordance with the standard protocol provided by the manufacturers of the sequencer. The bioinformatic analysis included five major steps: raw data quality control, genome assembly, genome component analysis, functional annotation, and genome visualization. In brief, the quality control of raw short reads from Illumina sequencing and raw long reads from Nanopore sequencing were performed in Fastp 0.19.5 and Mecat 2, respectively. Then, the clean short reads and long reads were co-assembled to reconstruct complete genomes using Unicycle (https://github.com/rrwick/Unicycler) to generate complete sequences. The coding sequences were predicted using Glimmer version 3.02[59]. Databases such as KEGG, COG, GO, and CAZy were used for functional annotation. In addition, MUMmer software (v3.23) was used to compare the target genome with the reference genome to determine the collinearity between the genomes.

## Proteomic analysis of PVC film degradation by strain EMBL-1

**Experimental setups for proteomic analysis.** To mine enzymatic activities and metabolic pathways related to PVC degradation, biodegradation of PVC film by the strain EMBL-1 was conducted with and without an additional supply of 1% ($w/v$) glucose (to distinguish the proteomic signals of the strain from that of PVC film). Then, both intracellular and extracellular proteins were separately extracted from the cells harvested after 30 days based on the acetone precipitation method. The ability of the protein solutions to degrade PVC film was further tested in vitro. The following experiments were set up in nine replicates: (1) MSM liquid medium + EMBL-1 (OD600 = 0.2) + PVC film (weighed), (2) MSM liquid medium + EMBL-1 (OD600 = 0.2) + PVC film (weighed) + glucose (1%, $w/v$), (3) MSM liquid medium + PVC film (weighed), (4) MSM liquid medium + PVC film (weighed) + glucose (1%, $w/v$). All treatments were cultured in a shake (150 rpm, 30 °C) for 30 days and PVC films were recovered and weighed every 10 days. The pretreatment and post-treatment methods of PVC film were the same as described as above. After the experiment, the cell culture solution in each group was collected for the next steps.

**Extraction of intracellular and extracellular proteins.** The collected cells of strain EMBL-1 in two groups were lysed by using Branson SFX550 Ultrasonicator, and then the solution was centrifuged to obtain supernatant for further operation. The collected culture solution in two groups were concentrated 10 times for next step. Acetone precipitation method was used to extract intracellular (IN) and extracellular (OUT) proteins. Specific steps were described as follows: a mixture of extraction solution and pre-cooled acetone ($v:v$ = 1:1) was stirring for 1 h at 0 °C, and then placed on 4 °C overnight. The mixture in each group was concentrated (10000 rpm, 4 °C) to harvest the proteins from four groups: IN (intracellular protein in group 1), OUT (extracellular protein in group 1), INglu (intracellular protein in group 2), and OUTglu (extracellular protein in group 2).

**Test of PVC-degradation activity of protein extracts.** PBS solution was used to dissolve the above protein extracts. The experiments were set up in triplicate as follows: (1) PBS solution + IN (0.1 mg/mL) + PVC film (weighed), (2) PBS solution + OUT (0.1 mg/mL) + PVC film (weighed), (3) PBS solution + Inglu (0.1 mg/mL) + PVC film (weighed), (4) PBS solution + OUTglu (0.1 mg/mL) + PVC film (weighed), and (5) PBS solution + PVC film (weighed). All treatments were cultured in a shake (150 rpm, 30 °C) for 48 h before the PVC films were recovered and weighed in each treatment. The weight loss was used to evaluate the PVC-degradation activity of proteins in the four groups: IN, OUT, Inglu, and OUTglu.

**Mass spectrometry analysis of proteome.** Proteins were resolved with a Thermo Ultimate 3000 integrated nano-HPLC system that directly interfaced with a Thermo Orbitrap Fusion Lumos mass spectrometer (LC-MS/MS) to explore some related PVC degradation proteins. The sodium dodecyl-sulfate polyacrylamide gel electrophoresis (SDS-PAGE) was used to separate the protein extracts and stained with Coomassie Blue G-250. The gel bands of target were cut into pieces. Sample was digested by trypsin with prior reduction and alkylation in 50 mM ammonium bicarbonate at 37 °C overnight. The digested products were extracted twice with 1% formic acid in 50% acetonitrile aqueous solution and dried to reduce volume by speed Vacuum Concentrator. The peptides were separated by a 65 min gradient elution at a flow rate 0.300 μL/min with the Thermo Ultimate 3000 integrated nano-HPLC system which is directly interfaced with the Thermo orbitrap fusion lumos mass spectrometer. The analytical column was a home-made fused silica capillary column (75 μm ID, 150 mm length; Upchurch, Oak Harbor, WA) packed with C-18 resin (300 A, 3 μm, Varian, Lexington, MA). Mobile phase A consisted of 0.1% formic acid, and mobile phase B consisted of 80% acetonitrile and 0.1% formic acid.

The mass spectrometer was operated in the data-dependent acquisition mode using the Xcalibur 4.1 software and there is a single full-scan mass spectrum in the Orbitrap (300–1800 m/z, 60,000 resolution) followed by 20 data-dependent MS/MS scans at 30% normalized collision energy. Each mass spectrum was analyzed using the Peak studio for the database searching. The reference strain is *Klebsiella variicola* (strain 118). Based on the PVC-degradation activity of four proteins (OUT > OUTglu > IN > INglu), statistical analysis was focused on the proteins shared by the OUT group and the IN group which showed degrading activities of PVC film. In total, 39 proteins were firstly selected within the yellow dotted line. Further, the relative abundance of these proteins in their respective groups was calculated and visualized in a heatmap. Moreover, the differential profiles in protein expression between the two groups were calculated based on the relative abundance to further narrow the list of potential PVC-degrading proteins (Log2(OUT/IN) ≥ 3). The mass spectrometry proteomics data have been deposited to the ProteomeXchange Consortium via the PRIDE partner repository with the dataset identifier PXD035850.

## Transcriptomic analysis of PVC film degradation by strain EMBL-1

**Experimental design.** To mine genes related to PVC degradation, degradation of PVC film by the strain EMBL-1 was conducted for 10 days with (a) MSM medium + EMBL-1 ($OD_{600}$ = 0.2) and (b) MSM medium +EMBL-1 ($OD_{600}$ = 0.2) + PVC film (weighed), with three repeats per group. All the treatment groups were cultured in a shaker (30 °C, 150 rpm). By the end of the experiment, the liquid culture was centrifuged at 4 °C to harvest the cells. The total RNA from each treatment group was extracted, and RNA integrity was assessed using the RNA Nano 6000 Assay Kit for the Agilent Bioanalyzer 2100 system (Agilent Technologies). Sequencing libraries were generated using the NEBNext Ultra Directional RNA Library Prep Kit for Illumina (NEB) and sequenced on the Illumina HiSeq2500 platform. Sequencing was performed at Beijing Novogene Bioinformatics Technology Co., Ltd. The raw transcriptomic data were uploaded to the NCBI database under accession code PRJNA866083.

**Bioinformatic analysis.** Raw reads were first filtered using fastp to remove the reads that contained 10 low-quality bases (base quality score <20) or lengths shorter than 36 bp. Then, the resulting HQ reads were aligned to the *K. variicola* reference genome (*K. variicola* strain FH-1) using hisat2. After alignment, the read counts for each gene were extracted using htseq-count. The gene expression profiles of triplicate transcriptomes in the two groups were compared with PCoA, which inspected one outlier dataset in each group (due to unexpected experimental errors) that was discarded from downstream analysis. DE at the gene level in our two groups (group a and group b) was evaluated using edgeR version 3.30.3, implemented in R 4.0.3. The $p$ values presented were adjusted for multiple testing with the procedure of Benjamini and Hochberg to control the type I error rate, and a cutoff of $p \leq 0.05$ was used as a threshold to define DE. Kraken2 was used to check for contamination in the RNA-seq data.

## Functional verification and degradation byproducts of catalase-peroxidase

Catalase-peroxidase, both previously known to degrade polymers (e.g., lignin[39]) and found in our study as the 4th most highly expressed extracellular protein during the PVC-dependent growth of strain EMBL-1 (Fig. 4d), was assumed to play a role in depolymerization of PVC. In order to verify this hypothesis, gene encoding catalase-peroxidase was identified from the genome of strain EMBL-1, and expressed and purified the enzyme in vitro by prokaryotic expression and purification method.

**Prokaryotic expression and purification**. The gene sequence of catalase-peroxidase was found from the genome information of strain EMBL-1. Appropriate restriction sites were added to the designed primers (F: 5′-ACGAATTCATGAGCACGTCTAACGAC-3′, R: 5′-AACTC-GAGCAGGTCGAAGCGGTCGA-3′, the bold font showed the restriction endonuclease sites EcoR and XhoI) designed to correspond to Cp. Primer synthesis and DNA sequencing services were provided by Shanghai Bioengineering Co., Ltd. (Shanghai, China). Using procedures developed earlier[60], the expression vector was constructed and expressed in vitro and purified. *E. coli* (DE3) cells containing the constructed vectors were inoculated into fresh LB medium containing kanamycin (0.5 mg/mL) and incubated at 37 °C in a rotary shaker at 150 rpm until reaching an $OD_{600}$ of 0.6. The recombinant strains were then induced with 0.6 mM isopropyl-b-D-thiogalactopyranoside at 37 °C for 2 h. The cells were collected by centrifugation at 6000 rpm for 5 min, and target protein expression was verified by 10% SDS-PAGE. The cells were resuspended in buffer A (20 mM Tris-HCl, 300 mM NaCl, 0.1% Triton-100, pH 8.0) in an ice bath, lysed by ultrasonication, and centrifugation at 12,000 rpm for 20 min. The supernatant was purified using an Ni-IDA agarose magnetic beads (Beijing Biomed Co., Ltd.) as per the manufacturer's instructions. The eluate was dialyzed against 20 mM Tris, 50 mM NaCl, pH 8.0, and the purity of the recombinant proteins was determined by 10% SDS-PAGE. Protein concentrations were determined by the Bradford method.

**Determination of catalase-peroxidase activity**. The activities of peroxidase (POD) and catalase (CAT) were determined by using kits purchased from Beijing Solarbio Science (China). All measurements were performed according to the manufacturer's instructions. Catalase (CAT) activity determination: $H_2O_2$ has a characteristic absorption peak at 240 nm. CAT can decompose $H_2O_2$, so that the absorbance of the reaction solution at 240 nm decreases with the reaction time. CAT activity can be calculated according to the change rate of absorbance. Definition of unit: every mg of protein catalyzes 1 per minute in the reaction system μ moL $H_2O_2$ degradation is defined as an enzyme activity unit. Peroxidase (POD) activity determination: POD catalyzes $H_2O_2$ to oxidize specific substrates and has characteristic light absorption at 470 nm. Definition of unit: 0.01 change in A470 per minute per mg of protein per ml of reaction system is an enzyme activity unit.

**Preparation of pure PVC polymer**. To design an assay for detecting the depolymerization activity of the catalase-peroxidase, pure PVC was prepared using PVC film with an extraction method inspired by the national standard method (GB/T 39110-2020). Specific steps are described as follows: 0.1 g of the film was added to 30 mL of THF solution and shaken to dissolve into a transparent solution. 70 mL of methanol solution was slowly added to the above solution and stirred well. During this process, a large amount of PVC polymer precipitated, and then the supernatant solution was discarded after standing for 1 h. The precipitates were collected and washed sequentially with 70 mL of methanol, 70 mL of ethanol and 70 mL of n-hexane solution to remove residual additives. Finally, the cleaned precipitates were collected and dried at 45 °C for 24 h, which was pure PVC powder. To check whether there is a difference in molecular weight between PVC powder and PVC film, a molecular weight detection was performed on the PVC powder. The results showed that the molecular weight of the PVC powder (Mn = 90.1 ± 0.38 KD, Mw = 165.71 ± 2.67 KD) was the same as that of the PVC film (Mn = 90.0 ± 0.31 KD, Mw = 163.49 ± 0.93 KD), which suggested that the powder could meet the basic requirements of the experiment. The experimental design was as follows: (1) Reaction Buffer (50 mM Tris, 300 mM NaCl, 1 mL) + catalase-peroxidase (50 μg/mL), (2) Reaction Buffer (50 mM Tris, 300 mM NaCl, 1 mL) + catalase-peroxidase (50 μg/mL) + PVC (100 mg), (3) Reaction Buffer (50 mM Tris, 300 mM NaCl, 1 mL) + PVC (100 mg). All treatments were repeated

three times and cultured in a shake (150 rpm, 30 °C) for 96 h. To validate and follow PVC biodegradation by catalase-peroxidase, FTIR and APC were used together to analyze the temporal changes in the physiochemical properties following the methods as described above.

**Characterization of PVC biodegradation products**. To obtain more evidence of the biodegradation of PVC by catalase-peroxidase, GC-MS was used to detect potential biodegradation products of PVC in the reaction solutions. Reaction solution (1 mL) was centrifuged (10,000×*g*, 10 min) to collect the supernatant. The supernatant was freeze-dried and re-dissolved in 1 mL dichloromethane, then 2 μL filtered supernatant was used for GC-MS analysis performed on GC-MS system equipped with a TG-5 ms (30 m long, 0.25 mm internal diameter and 0.25 μm thickness) chromatographic column. The injection-port was set at 300 °C. During operation the column temperature was held for 4 min at 50 °C, then raised to 300 °C at 20 °C rise per min, and finally, held for 15 min at 300 °C. The flow rate was set at 1 mL/min. Helium was used as a carrier gas. Ions/fragments were monitored in scanning mode through 50–600 Amu[61]. Some potential degradation products were identified according to the high match score (>800) of each compound in the NIST library.

**Multiomic prediction of degradation pathway of PVC film**
To further explore the mechanism underlying the degradation of PVC film by strain EMBL-1, results from multiomic analyses, i.e., genomic, transcriptomic, proteomic, and metabolite analyses, were used together to propose a putative pathway for PVC degradation. In brief, the potential plastic-degrading genes encoded in the EMBL-1 genome (Table 1 and Dataset S1) and the metabolites detected by GC-MS were used together to build a putative PVC degradation pathway. Furthermore, 39 proteins jointly expressed during PVC-dependent growth of strain EMBL-1 (Fig. 4d and Dataset S2) were aligned against the 96 differentially expressed genes revealed by transcriptomic analysis (Fig. 4f and Dataset S3) using NCBI's BLAST + 2.9.0 at an e value cutoff of 0.01, generating a list of gene expression and proteomic changes ascribed to the PVC-dependent metabolism of the strain.

**Reporting summary**
Further information on research design is available in the Nature Research Reporting Summary linked to this article.

## Data availability
The 16 S rRNA gene amplicon sequence, complete genome sequence, and raw transcriptomic data of strain EMBL-1 generated in this study have been deposited in the NCBI database under accession code MZ475068, CP079802, and PRJNA866083, respectively. The mass spectrometry proteomics data have been deposited to the ProteomeXchange Consortium via the PRIDE partner repository with the dataset identifier PXD035850. Source data are provided with this paper.

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

## Acknowledgements

This work was supported by Zhejiang Provincial Natural Science Foundation of China (grant no. LR22D010001 to J.F.), National Natural Science Foundation of China (grant no.: 32100091 to Z.Z. and 51908467 to J.F.), Research Center for Industries of the Future (grant no. WU2022C030 to J.F.), and Center of Synthetic Biology and Integrated Bioengineering. We would like to thank Dr. Lihan Zhang and Dr. Yubo Wang at Westlake University for helpful discussion and Professor Ningyi Zhou at Shanghai Jiao Tong University for helpful discussion and suggestions. Thank Dr. Yu Xia for the advice on whole-genome sequencing. We also thank Dr. Xingyu Lu, Dr. Xiaohuo Shi, Dr. Yinjuan Chen, Yu Huang, and Ke Wang from the Instrumentation and Service Center for Molecular Sciences and Dr. Xiuxiu Zhao, Xue Bai, and Jinheng Pan from the Biomedical Research Core Facilities at Westlake University for assistance and discussion during the experiment. Thank the Westlake University HPC Center for computation support.

## Author contributions

F.J. and Z.Z. designed the project and Z.Z. performed most of the experiments. D.C.Y. performed some experiments on the breeding of *Spodoptera frugiperda* larva. J.L.Z. provided the place for breeding *Spodoptera frugiperda* larva and provided technical guidance. H.R.P. was responsible for the analysis of sequencing data under the supervision of F.J. and with technical help from G.Q.Z. Z.Z. and F.J. co-wrote the manuscript. F.J. supervised experiment design, supervised data analysis, provided funding, and draft the manuscript.

## Competing interests

The authors declare no competing interests.
