## [Peer Review File · Nature Communications]

Reviewers' Comments:

Reviewer #1:

Remarks to the Author:

Microbial or enzymatic degradation of plastic polymers is a currently intensively explored research area. Whereas for plastics containing readily hydrolysable bonds such as in polyethylene terephthalate (PET), a range of lipase/cutinase/esterase have been discovered and optimized in the past decade, for many other plastics (like polyethylene or polystyrene) such a degradation is much more difficult. Only a few studies so far could provide sufficient scientific evidence that indeed the polymer is converted by enzymatic steps with indications about the reaction steps taking place.

One of the very challenging polymers is PVC, polyvinyl chloride, and in this manuscript Zhe et al. describe very interesting results for its degradation. Initially, they have observed that the larvae of *S. frugiperda* can act on PVC and concluded that its intestinal microbiota should contain microbes / enzymes able to act on the highly recalcitrant PVC. They then performed selective enrichment and identified the biofilm-forming strain *Klebsiella* sp. EMBL-1. They could show that this microbe can indeed degrade PVC supported by physicochemical and morphology studies followed by a range of 'omics technologies which finally led to the identification of enzyme encoding genes. Based on this data they could propose a convincing biodegradation pathway.

This is very important, solid and innovative study, which should be published after the comments given below have been addressed in a revision.

Terminology: they should not write about 'recycling' of PVC as the building blocks to make new PVC (vinylchloride) are not accessible after degradation (in contrast to terephthalic acid from PET, which can be used to indeed recycle the PET without using petrol again). Hence, they only should write about degradation and maybe 'upcycling', if the products released from degradation can be used for alternative chemicals or just cell growth/biomass production. Degradation of PVC is also not contributing (in a strict sense) to a circular bioeconomy.

The degradation pathway proposed is interesting from an evolutionary point of view (as the 'man-made' plastics are out in nature since less than a 100 years). Connected to the discovery by Yoshida et al. for the first PET degrading and assimilating microorganism *Iodanella sakaiensis*, a perspectives article published in *Science* points this evolutionary aspect out and hence that article could also be cited and discussed here: Bornscheuer, U.T. (2016), Feeding on plastic, *Science*, 351, 1155-1156

p.4, line 74: ... Nazia Khatoon's documentation... better write: "the documentation reported by Khatoon et al..."

p.5, line 115: ... on PVC film was lower than that...: please provide experimental data (xx % lower) so that the reader learns what "was lower than" means.

p.6, line 135: "... can lead" must read "...must lead"

p.8, line 193:

... microbial oxidation: please explain how an oxidation can take place – assuming that oxygen is needed for this – in the rather anaerobic intestinal microbiota? They could find genes annotated as laccase, alkane monooxygenase, peroxidase, which all need oxygen.

The most interesting and still open question is how the dichlorination takes place. This must be a reaction unique for PVC as other recalcitrant polymers like PE, PP, PS do not contain chlorine/halogens. From literature, many dehalogenases are known, which have been for instance explored for dehalogenation of toxic pesticides. I recommend to check literature in that field (key scientists: Jiri Damborsky, Masaryk University, Brno, Czech Republic and Dick Janssen, Univ. of Groningen, Groningen, NL) and to also explore the genome for dehalogenases (some dehalogenases are related to epoxide hydrolases, so maybe there could be a promiscuous epoxide hydrolase evolved with dehalogenase activity?)

p.9, lines 223-229: it should be made clearer/discussed, which of the compounds indeed can be degradation products from PVC or originate from additives/plasticizers (or hopefully not from wrong assignments of compounds in the MS database).

p.12, line 289: I recommend to name the enzymes identified (catalase-peroxidase, enolase, aldehyde dehydrogenase, dehydratase etc.) in the abstract where they mention the biodegradation

pathway. It would be very helpful for the reader to learn this finding in the abstract already.
p.16, line 404: as above, write "Gravouil et al." not "Eyheraugibel".

Figure 5: please improve the chemical structures layout a bit. In some compounds the "OH" group is not exactly located at the bond or the "O" in the carboxylic acid. The green boxes are too dark and hence the text inside can not be read Use lighter green or white text font.

Reviewer #2:

Remarks to the Author:

The authors discovered that the larvae *Spodoptera frugiperda* feed and survive on polyvinyl chloride (PVC) film, and successfully isolated a bacterium *Klebsiella* sp. EMBL-1 strain that degrades and utilizes PVC from the larval intestine. Although this is a similar report with that of PE-feeding larva and its intestine bacteria capable of degrading PE by Yang et al. (*Environmental Science and Technology*, 2014, 13776-13784, 48(23), not referenced in this manuscript), it is meaningful that this interesting scheme was validated in the PVC biodegradation. However, it is a pity that the multi-omics analysis in this study did not reach the identification of enzymes involved in the PVC degradation.

Specific comments

Line 116, Contrast between the feces is unclear. What is the point?

Line 126, What is the authors' expectation for gentamicin?

Line 135-, Missing insight into the changes in microbiota composition.

Line 165, Need explanation about the growth study using soybean oil as a carbon source.

Line 260, PET is not the subject of this study.

Fig. 5., Too speculative. Functional analysis of important enzymes (at least) involved in the predicted pathway is required for suggesting this mechanism.

Reviewer #3:

Remarks to the Author:

The manuscript by Zhe et al., describes the findings on the biodegradation of PVC by the intestinal microbiota of the larva of *S. frugiperda*, specially by a specific bacterial isolate. The authors have used a combination of transcriptomics, proteomics, metabolomics, as well as analytical chemistry to elucidate the PVC degradation pathway by this microorganism. In previous publications on PVC degradation, it was revealed that the microorganisms consumed the plasticizers and not the PVC itself. In this manuscript, the authors have demonstrated that their strain does not consume the main additives that were present in the PVC film, but the O.D. increases when the PVC film is supplied, which is given as an evidence that the bacterial strain can use PVC as a carbon source.

At the moment, a couple of problems prevent me from reviewing the manuscript to its full capacity:

a. The figures 2a and 2b have a very low resolution and are very small, making it very hard to distinguish any details. Especially the negative control in 2b is so small that it is not even possible to read the scale.

b. Figure 1 similarly is hard to interpret. The numbering/lettering of the figures do not match between the figure caption and the figure. The panels 1 and 2 show the model organism on corn leaves and PVC, 3 shows corn leaves in petri dishes, neither of which are very informative. Panel 4 also shows petri dishes with a film in the foreground, but it is hard to distinguish what it is as the figure is too small, and what it is has not been explained in the caption.

c. Figure 2 e depicts the day 0 control, but not the day 90 abiotic (no bacteria) control. This should be included.

Other remarks:

1. Lines 56-60: I do not agree with the transition from "no sustainable approach available for disposal of PVC wastes" to "biological treatment of PVC." Finding a PVC-degrading microorganisms will not solve the problem of improper plastic waste disposal, and there is little we can do for recycling either way unless we properly collect and process the waste. The authors should discuss

the current options for PVC recycling, and how biological recycling would be superior.

2. Line 117: Couldn't the surface changes etc be a consequence of chewing? A control with the excreted PVC pieces in the feces of antibiotic-treated larvae should be shown as comparison.

3. Lines 127-134: Have the authors employed a gentamycin+corn control? I assume that the larvae have a very limited digestive system, and depend on their intestinal microbiota also for their natural substrates. Does the gentamycin treatment have any other toxicity on the larvae other than affecting their intestinal microbiota?

4. Line 168: Figure S3b shows a difference of the contact angle from between control and EMBL-1 to be about 10°. Is this difference significant? Authors claim that it is, but as far as I can see, no statistical test has been applied. Similarly, the values between S3e and S3f look very similar. Are these differences significant?

5. Line 228: Why weren't the degradation/growth experiments conducted with additive-free PVC? Is this material not available?

6. Line 268: As far as I can understand, PVC+glucose was used as a control to PVC only. Wouldn't it have been a better option to use PVC only vs glucose only? What was the reasoning behind this control?

7. Line 272: Wouldn't it be expected that increased microbial biomass leads to more degradation/weight loss? What do the authors propose as the reason behind the lack of differences between PVC vs. PVC+glucose?

8. Line 272: Was a no-bacteria control employed? Was any weight-loss observed? I am especially asking because the material used contains a lot of additives, and although it was pretreated, the leakage of remaining additives can also cause weight loss. This data should be shown as well.

9. Line 343: Since the protein was not recombinantly expressed and characterized, it should be stated that this is a hypothesis. We do not know if this protein degrades PVC.

10. Line 347: The authors describe that laccase, monooxygenase, and dioxygenases process the degradation products, but then go on to state that these enzymes were not detected in the proteome on line 353, which indicates that the former claim is not supported. Were these genes upregulated in the transcriptome?

11. Line 366: Is there a chance that the larvae could be using the leftover soybean oil in PVC for survival?

12. Generally, the discussion part is inadequate and repeats the results part. Again, the authors mention mRNA, but I did not find a report of transcriptome results or the consolidation of these with the proteome results. The conclusion part is also repetitive. Especially lines 412-422 are repetitions of the discussion part.

13. Figure 3: Panel b is not very informative for the purposes of this paper, and quite hard to distinguish the colors. Removing this panel and extending tree with more species and meta information (where they have been isolated etc) would make the figure more informative.

14. Figure 4: The proteomic and transcriptomic data should be presented comparatively and consolidated (e.g comparison of fold changes between transcriptome and proteome for genes of interest). At the moment it is hard to draw conclusions. What is the difference between the heatmap and the bar chart in terms of what they depict?

15. Figures S3d & S6f, and Figure 4e: It seems like the initial O.D. of the cultures is very high, they were already inoculated at 0.2-0.3. Why was this low dilution factor used? This would lead to the carry over of a high number of late-growth-stage bacteria. Also, for the experiment where the growth in soybean oil was assessed, the starting O.D. of the experiment and the control are different.

Reviewer #1:

Microbial or enzymatic degradation of plastic polymers is a currently intensively explored research area. Whereas for plastics containing readily hydrolysable bonds such as in polyethylene terephthalate (PET), a range of lipase/cutinase/esterase have been discovered and optimized in the past decade, for many other plastics (like polyethylene or polystyrene) such a degradation is much more difficult. Only a few studies so far could provide sufficient scientific evidence that indeed the polymer is converted by enzymatic steps with indications about the reaction steps taking place.

One of the very challenging polymers is PVC, polyvinyl chloride, and in this manuscript Zhe et al. describe very interesting results for its degradation. Initially, they have observed that the larvae of *S. frugiperda* can act on PVC and concluded that its intestinal microbiota should contain microbes / enzymes able to act on the highly recalcitrant PVC. They then performed selective enrichment and identified the biofilm-forming strain *Klebsiella* sp. EMBL-1. They could show that this microbe can indeed degrade PVC supported by physicochemical and morphology studies followed by a range of 'omics technologies which finally led to the identification of enzyme encoding genes. Based on this data they could propose a convincing biodegradation pathway.

This is very important, solid and innovative study, which should be published after the comments given below have been addressed in a revision.

Response: We sincerely appreciate you for your time in reviewing the manuscript and giving your valuable comments. We have thoroughly considered your precious comments and suggestions, and carefully addressed every one of them as best as we can. Revisions in the manuscript were highlighted in red. Here, we provide a point-to-point response to your major and specific comments below.

Terminology: they should not write about 'recycling' of PVC as the building blocks to make new PVC (vinylchloride) are not accessible after degradation (in contrast to terephthalic acid from PET, which can be used to indeed recycle the PET without using petrol again). Hence, they only should write about degradation and maybe 'upcycling', if the products released from

degradation can be used for alternative chemicals or just cell growth/biomass production. Degradation of PVC is also not contributing (in a strict sense) to a circular bioeconomy.

Responses: Thanks for raising the important comments. We have revised ‘recycling’ as ‘upcycling’ according to your suggestion (Line 65 and Line 84). Meanwhile, we have also re-written about biological degradation and upcycling and avoided improper assignment of PVC degradation to a circular bioeconomy, as suggested. The revised contents (Line 60-68) are extracted as follows to facilitate your further review.

Line 60-68: “Biological degradation and upcycling of plastic wastes are a promising approach for the future sustainable development of a green bioeconomy. In order to realize biological plastics upcycling, on the one hand, more plastic biodegradation strains and enzymes should be developed, and on the other hand, the intermediate biodegradation products should be recovered for alternative chemicals or biomass production ⁶.”

The degradation pathway proposed is interesting from an evolutionary point of view (as the ‘man-made’ plastics are out in nature since less than a 100 years). Connected to the discovery by Yoshida et al. for the first PET degrading and assimilating microorganism *Iodanella sakaiensis*, a perspectives article published in *Science* points this evolutionary aspect out and hence that article could also be cited and discussed here: Bornscheuer, U.T. (2016), Feeding on plastic, *Science*, 351, 1155-1156

Responses: Thanks for the insightful comments. Following your suggestion, we have added citation to this key reference (Bornscheuer, U.T. (2016), Feeding on plastic, *Science*, 351, 1155-1156) (Line 68 and Line 432) and discussed on the evolutionary aspect of plastic-degrading enzymes and pathways (Line 427-439). Moreover, we have also added experiments to functionally verify catalase-peroxidase on its the activity of PVC depolymerization. This enzyme was previously reported to have lignin degradation activity. To facilitate your review, the updated contents (Line 427-439) are extracted as follows to facilitate your further review.

Line 427-439: “In this study, we reported for the first time experimentally verified activities of catalase-peroxidase of strain EMBL-1 in PVC depolymerization. This new finding raises

intriguing questions on the evolutionary principles and perspectives of PVC-degrading enzymes and pathways, as insightfully conceived for PET biodegradation by Bornscheuer (2016) ⁶. Plastic-degrading enzymes should have been evolved more easily from existing homologous enzymes that empower degrading microbes and even their hosts to better adapt to additional nutrient niches (e.g., plastic feeding by *S. frugiperda* larvae and intestinal strain EMBL-1 first demonstrated here) and environmental changes (e.g., microplastic pollution). Based on this view, the PVC-depolymerization capacity and activities of catalase-peroxidase enzyme discovered in strain EMBL-1 are most likely evolutionarily linked to those of other polymers, such as lignin ³⁹.”

p.4, line 74: ... Nazia Khatoon’s documentation... better write: “the documentation reported by Khatoon et al...”

Responses: We appreciate your comment and revise as suggested. In Line 80, “... Nazia Khatoon’s documentation...” is rephrased as “...the documentation reported by Khatoon et al...”.

p.5, line 115: ... on PVC film was lower than that...: please provide experimental data (xx % lower) so that the reader learns what “was lower than” means.

Responses: Thanks and revised as suggested. We have provided the experimental data in the Line 120 of the revised manuscript.

p.6, line 135: “... can lead” must read “...must lead”

Responses: Thanks and revised as suggested. In Line 145, ‘...can lead’ is replaced by ‘...must lead’.

p.8, line 193:

... microbial oxidation: please explain how an oxidation can take place – assuming that oxygen is needed for this – in the rather anaerobic intestinal microbiota? They could find genes annotated as laccase, alkane monooxygenase, peroxidase, which all need oxygen.

Responses: Thanks for the question. The degradation experiments of the strain EMBL-1 reported here were all based on in vitro experiments on the isolate, not in the intestine of larva.

To make this point clearer and avoid potential reader confusion, we have rephrased the short sentence as ‘...the strain can degrade the PVC film via oxidation’ (Line 208).

The most interesting and still open question is how the dichlorination takes place. This must be a reaction unique for PVC as other recalcitrant polymers like PE, PP, PS do not contain chlorine/halogens. From literature, many dehalogenases are known, which have been for instance explored for dehalogenation of toxic pesticides. I recommend to check literature in that field (key scientists: Jiri Damborsky, Masaryk University, Brno, Czech Republic and Dick Janssen, Univ. of Groningen, Groningen, NL) and to also explore the genome for dehalogenases (some dehalogenases are related to epoxide hydrolases, so maybe there could be a promiscuous epoxide hydrolase been evolved with dehalogenase activity?)

Responses: The reviewer’s insightful suggestion on dehalogenase is very much appreciated. Enlightened by your suggestion, we searched the relevant literature on dehalogenase. Finally, we re-searched our functional gene annotation results of strain EMBL-1 and identified at least two groups of genes encoding potential dehalogenases from the strain genome, such as non-heme chloroperoxidase and HAD family hydrolase genes which are reported in the articles of Dick Janssen [1] and Jiri Damborsky [2] to be involved in dichlorination of some halogenated compounds. These genes are proposed promiscuous dehalogenase genes in the strain EMBL-1. We added some discussion on the PVC dechlorination in the revised manuscript (Line 439-444). To facilitate your further review, the contents are extracted as follows:

Line 439-444: “we identified at least two groups of genes potentially encoding dehalogenases in the genome of strain EMBL-1, i.e., non-heme chloroperoxidase and HAD family hydrolase genes which are reported to trigger oxidative dechlorination activities of some halogenated compounds^{53,54} (Dataset S1). The chloroperoxidase gene of the strain is more likely to be involved in PVC dechlorination evident in the NMR results.”

Reference:

[1] Atashgahi S, Liebensteiner M G, Janssen D B, et al. *Microbial Synthesis and Transformation of Inorganic and Organic Chlorine Compounds*[J]. *Frontiers in Microbiology*, 2018, 9.

[2] Schuiten, E. D., Badenhorst, C. P. S., Palm, G. J., Berndt, L., Lammers, M., Mican, J., Bednar, D., Damborsky, J., Bornscheuer, U. T., 2021: Promiscuous Dehalogenase Activity of the Epoxide Hydrolase CorEH from *Corynebacterium* sp. C12. *ACS Catalysis* 11: 6113-6120.

p.9, lines 223-229: it should be made clearer/discussed, which of the compounds indeed can be degradation products from PVC or originate from additives/plasticizers (or hopefully not from wrong assignments of compounds in the MS database).

Responses: Thank you for your comments. First of all, we used a commercial PVC film in this study, which was identified by TD/Py-GC-MS analysis to contain three main additives (See Method S6), namely dioctyl adipate (DOA), dioctyl terephthalate (DOTP) and erucylamide. The results of degradation experiments (see Method S7) showed that strain EMBL-1 had no degradation activity to the three main additives (Line 244-246), which exclude the presence of byproducts of any main plasticizer. By careful comparative inspection of the GC-MS peaks with significant differences between the EMBL-1 group and the control group, compounds 1-5 were identified as the main degradation products of PVC.. Their structure were carefully confirmed according to the NIST library (high match score > 800, Line 240) to avoid mis-assignment of compounds in the MS database. Following your comments, we have slightly rephrased the relevant contents to make them clearer (Line 241-251).

p.12, line 289: I recommend to name the enzymes identified (catalase-peroxidase, enolase, aldehyde dehydrogenase, dehydratase etc.) in the abstract where they mention the biodegradation pathway. It would be very helpful for the reader to learn this finding in the abstract already.

Responses: Thanks for the helpful suggestion. We have named the identified enzymes in the updated abstract (Line 34-Line 36), as you recommended, to better inform readers on the key results of the biodegradation pathway.

p.16, line 404: as above, write “Gravouil et al.” not “Eyheraugibel”.

Responses: We appreciate your comment and revise as suggested (Line 454).

Figure 5: please improve the chemical structures layout a bit. In some compounds the “OH” group is not exactly located at the bond or the “O” in the carboxylic acid. The green boxes are too dark and hence the text inside can not be read Use lighter green or white text font.

Responses: Fully agreed and revised as suggested. We have improved the chemical structures layout and changed the black text font into white text font to make it clearer.

Reviewer #2 (Remarks to the Author):

The authors discovered that the larvae *Spodoptera frugiperda* feed and survive on polyvinyl chloride (PVC) film, and successfully isolated a bacterium *Klebsiella* sp. EMBL-1 strain that degrades and utilizes PVC from the larval intestine. Although this is a similar report with that of PE-feeding larva and its intestine bacteria capable of degrading PE by Yang et al. (Environmental Science and Technology, 2014, 13776-13784, 48(23), not referenced in this manuscript), it is meaningful that this interesting scheme was validated in the PVC biodegradation. However, it is a pity that the multi-omics analysis in this study did not reach the identification of enzymes involved in the PVC degradation.

Responses: We sincerely appreciate you for your time in reviewing the manuscript and giving your valuable comments. We have thoroughly considered your useful comments and thought-provoking questions, and carefully addressed every one of them as best as we can. Following your useful suggestion, we have added citation to the article by Yang et al. 2014, 48(23): 13776-13784 as a key reference (Line 69, Line 90 and Line 132), as suggested. Moreover, inspired by your comments on the identification of enzymes involved in PVC degradation, we finished new experiments (see Method S10 and Method S11) to functionally verify enzymes involved in the PVC degradation to make our research outcomes more complete. The major results and discussion are provided in Line 335-Line 351 and Line 658-Line 676 of the revised manuscript. In brief, we discovered and verified the catalase-peroxidase of strain EMBL-1 as a promiscuous enzyme involved in the depolymerization of PVC. Here, we provide a point-to-point response to your specific comments below with revisions highlighted in red in the revised manuscript.

Specific comments

Line 116, Contrast between the feces is unclear. What is the point?

Responses: We appreciate your comment and question. The main point here is to let readers visually see and intuitively compare the difference between the feces samples after eating corn leaves and eating PVC film. To avoid potential reader confusion and facilitate reading, we have deleted 'as also morphologically manifested by the contrasting excreted feces' from the revised manuscript.

Line 126, What is the authors' expectation for gentamicin?

Responses: Thanks for raising this important question. Based on Yang et al.'s research on *Tenebrio molitor* feeding on PE and PS, we expected that gentamicin can kill or inactivate most of the gut microbes, wanted to use this antibiotic to preliminarily explore whether the larva reply on gut microbes to degrade PVC films. We have rephrased the whole sentence to make this point clearer in the revised manuscript (Line 120-127 and Line 132-135).

Line 135-, Missing insight into the changes in microbiota composition.

Responses: We appreciate your constructive comment. We have added some discussion and insights into changes in microbial composition to this section. Please find the updated contents marked in red in Line 149- Line 152 and Line 161- Line 167 of the revised manuscript. The revised contents were extracted as follows to facilitate your further review.

Line 149-152: "...Proteobacteria-predominated microbiota ($87.5\pm 8.0\%$ to $49.5\pm 16.0\%$) to one co-dominated by Firmicutes ($11.9\pm 7.2\%$ to $44.2\pm 17.0\%$) (Figure 1c), revealing a close interconnection between Firmicutes members and PVC film degradation."

Line 161-167: "In summary, PVC film feeding leads to a strong structural (i.e., diversity and composition) shift in the intestinal bacteriome from phylum down to genus and ASV (a proxy for species) levels. Moreover, the identification of selectively enriched bacterial genera (e.g., *Enterococcus* and *Klebsiella*) not only reveals their potential role in the biodegradation of PVC films, but also enlightens field researchers on the taxonomically oriented isolation and screening of novel PVC-degrading strains."

Line 165, Need explanation about the growth study using soybean oil as a carbon source.

Responses: Thanks for raising this important point. In the original Supplementary Method S5, we have conducted growth experiments of strain EMBL-1 using soybean oil as the sole carbon source, and the results showed that the strain cannot utilize soybean oil for growth and reproduction. Following your helpful suggestion, we have optimized the original explanations on the growth study. The revisions were marked in red in Line 246-251, Line 557-561 of the revised version of main manuscript, and Line 138-139 and Line 297-299 of the revised version of Supplementary Information.

Line 260, PET is not the subject of this study.

Responses: We appreciate your comment and we have removed the content about PET as you suggested.

Fig. 5., Too speculative. Functional analysis of important enzymes (at least) involved in the predicted pathway is required for suggesting this mechanism.

Responses: Thanks for your comments and suggestion. We have strictly followed them to set up new experiments (see Method S10) and performed functional analysis of an important enzyme (catalase-peroxidase) involved in the predicted pathway, successfully verifying the degradation activity of this enzyme on the pure PVC polymer under in vitro conditions. Please find the updated contents and inspiring results in Line 335-351 and Line 658-676 of the revised manuscript. We appreciate deeply for raising this critical point on functional analysis, and we believe that the revisions made according to your important suggestion greatly improve the results.

Reviewer #3 (Remarks to the Author):

The manuscript by Zhe et al., describes the findings on the biodegradation of PVC by the intestinal microbiota of the larva of *S. frugiperda*, specially by a specific bacterial isolate. The authors have used a combination of transcriptomics, proteomics, metabolomics, as well as analytical chemistry to elucidate the PVC degradation pathway by this microorganism. In previous publications on PVC degradation, it was revealed that the microorganisms consumed

the plasticizers and not the PVC itself. In this manuscript, the authors have demonstrated that their strain does not consume the main additives that were present in the PVC film, but the O.D. increases when the PVC film is supplied, which is given as an evidence that the bacterial strain can use PVC as a carbon source.

Responses: We sincerely appreciate you for your time in reviewing our manuscript and offering your valuable comments and suggestions. We have thoroughly considered your concerns or comments and addressed every one of them as best as we can.

At the moment, a couple of problems prevent me from reviewing the manuscript to its full capacity:

a. The figures 2a and 2b have a very low resolution and are very small, making it very hard to distinguish any details. Especially the negative control in 2b is so small that it is not even possible to read the scale.

Responses: Thanks for your suggestion. Our sincere apologies for causing the inconvenience during your reading. In the revised version, we have updated with new clearer images and also enlarge its overall size.

b. Figure 1 similarly is hard to interpret. The numbering/lettering of the figures do not match between the figure caption and the figure. The panels 1 and 2 show the model organism on corn leaves and PVC, 3 shows corn leaves in petri dishes, neither of which are very informative. Panel 4 also shows petri dishes with a film in the foreground, but it is hard to distinguish what it is as the figure is too small, and what it is has not been explained in the caption.

Responses: Thanks for the useful comments for us to improve the figure style. Our sincere apologies for the missing explanation in the caption. In the revised version of the Figure 1, we have re-assembled the panels, added annotation to each panel, and expanded the figure caption with explanations for each group of panels. Please refer to the updated Figure 1a for the revisions according to your helpful comments.

c. Figure 2 e depicts the day 0 control, but not the day 90 abiotic (no bacteria) control. This should be included.

Responses: Thanks for raising this important point and careful check. Following your

suggestion, we have provided the supplementary data results of day 90 abiotic (no bacteria) control in the Figure S3g. In brief, the results showed that there is almost no change in the PVC film in the control group.

Other remarks:

1. Lines 56-60: I do not agree with the transition from “no sustainable approach available for disposal of PVC wastes” to “biological treatment of PVC.” Finding a PVC-degrading microorganisms will not solve the problem of improper plastic waste disposal, and there is little we can do for recycling either way unless we properly collect and process the waste. The authors should discuss the current options for PVC recycling, and how biological recycling would be superior.

Responses: Thanks for your insightful comments, which we fully agree. We have deleted the words of “no sustainable approach available for disposal of PVC wastes” and rephrased accordingly in a context of biological degradation of wasted plastics (Line 60-62). We have also added appropriate discussion on the biological degradation and upcycling of PVC (Line 63-68).

2. Line 117: Couldn't the surface changes etc be a consequence of chewing?

Responses: Thanks for raising this key point. Based on our experimental results provided in the main text and Figure S1, it is mainly a consequence of intestinal microbiota rather than chewing. To make this key point better delivered, we have rephrased the relevant contents and extracted them as follows to facilitate your further review.

Line 123-127: “Based on scanning electron microscopy (SEM) analysis (Figure S1), we found the PVC fragments recovered from excreted feces in the PVC group (a) showed strong surface damage in contrast with the Antibiotic group (b), revealing the importance of intestinal microbiota for PVC degradation.”

3. Lines 127-134: Have the authors employed a gentamycin+corn control?

Responses: Thank you for the question. During the laboratory experiments of this study, we

set up the control group in which gentamicin antibiotic was used to inactivate most intestinal microbes of the larva to resolve their importance for the degradation of PVC (instead of corn leaf). In other words, we specially employed a control treatment named as ‘Antibiotic group’ in which larva (n = 30 pcs) were fed with gentamicin-soaked corn leaf for 3 d, and then PVC film (Line 486-487). Therefore, we did not include a gentamicin+corn control because corn leaf degradation by intestinal microbes is not among the key considerations of our experimental design here, and we sincerely appreciated the thought-provoking question.

I assume that the larvae have a very limited digestive system, and depend on their intestinal microbiota also for their natural substrates. Does the gentamicin treatment have any other toxicity on the larvae other than affecting their intestinal microbiota?

Responses: Thanks for raising this important point. Based on the research by Yang et al. and Peng et al. and Brandon et al. on the feeding of *Tenebrio molitor* on PE [1] and PS [2,3], we directly used gentamicin as the antibiotic treatment group of *Spodoptera frugiperda* larva without experimental evaluation on the potential toxic effects of gentamicin on the larvae. We did not inspect identifiable toxic effects of gentamicin on larvae throughout the whole experiment.

Reference:

[1] Yang, J., Yang, Y., Wu, W. M., Zhao, J. & Jiang, L. Evidence of polyethylene biodegradation by bacterial strains from the guts of plastic-eating waxworms. *Environ Sci Technol* 48, 13776-13784, doi:10.1021/es504038a (2014).

[2] Peng, B. Y. et al. Biodegradation of Polystyrene by Dark (*Tenebrio obscurus*) and Yellow (*Tenebrio molitor*) Mealworms (Coleoptera: Tenebrionidae). *Environ Sci Technol* 53, 5256-5265, doi:10.1021/acs.est.8b06963 (2019).

[3] Brandon, A. M. et al. Biodegradation of Polyethylene and Plastic Mixtures in Mealworms (Larvae of *Tenebrio molitor*) and Effects on the Gut Microbiome. *Environ Sci Technol* 52, 6526-6533, doi:10.1021/acs.est.8b02301 (2018)

4. Line 168: Figure S3b shows a difference of the contact angle from between control and

EMBL-1 to be about 10°. Is this difference significant? Authors claim that it is, but as far as I can see, no statistical test has been applied.

Responses: Thanks for your important comment. The data results of contact angle from two groups were repeatable, based on triplicate experiments. Inspired your question, we performed a statistical test (*t*-test) of the two groups of samples (n= 3 each) and found significant difference in group means ($P<0.05$). We have updated the results in the caption of Figure S3 (Line 295-296, Supplementary Information).

Similarly, the values between S3e and S3f look very similar. Are these differences significant?

Responses: Thanks for your careful check and question. We have carefully check the results which showed that the T_{\max} of the films in the PVC group was obviously lower than that of the control group (279°C vs 310°C). To make this difference easier to visually inspect, we have slightly updated Figure S3e-S3f by adding arrow lines pointing to places with the greatest differences. Because the samples detection with the TGA was all conducted only once, it is infeasible to check the significance of these differences. Based on the empirical criterion employed by Lucia et al. about PVC degradation [1], we thought that the physical properties of PVC film in the strain EMBL-1 group had considerably changed.

Reference:

[1] *Giacomucci L et al. Polyvinyl chloride biodegradation by Pseudomonas citronellolis and Bacillus flexus. New biotechnology, 2019.*

5. Line 228: Why weren't the degradation/growth experiments conducted with additive-free PVC? Is this material not available?

Responses: Thanks for your comment and question. The initial findings of this study are originated from a serendipitous discovery on the larvae feeding on commercialized PVC films. The use of PVC film is more in line with both the original goals of our study and the practical situation in plastic waste, although additive-free PVC materials are also available. We have identified the additive composition of plastic film (Method S6 and Method S7), and the results showed that the film can meet our research needs.

6. Line 268: As far as I can understand, PVC+glucose was used as a control to PVC only. Wouldn't it have been a better option to use PVC only vs glucose only? What was the reasoning behind this control?

Responses: Thanks for raising the questions. The reasoning behind this control group in our experiments was to promote additional degradation activities of PVC via the addition of tiny amounts of easily-degradable glucose (1%). For example, microbial biodegradation of many other recalcitrant or toxic compounds, such as phenolic compounds, is known to be promoted when glucose is co-supplied. However, our results showed limited promotive effects of additional glucose on the PVC degradation.

7. Line 272: Wouldn't it be expected that increased microbial biomass leads to more degradation/weight loss? What do the authors propose as the reason behind the lack of differences between PVC vs. PVC+glucose?

Responses: Thanks for raising the questions. In theory, an increase in microbial biomass owing to additional glucose supply can lead to more degradation of PVC, as you pointed out. However, the rationale behind the setup of PVC+glucose group was to hopefully promote the PVC degradation activities so that we can better resolve the differential proteomic profiles, as we explained in your specific comment 6. The results showed limited or negligible difference between the PVC vs. PVC+glucose groups, suggesting no or limited promotion effects in the PVC biodegradation.

8. Line 272: Was a no-bacteria control employed? Was any weight-loss observed? I am especially asking because the material used contains a lot of additives, and although it was pretreated, the leakage of remaining additives can also cause weight loss. This data should be shown as well.

Responses: Thanks for raising these important points. Yes, a no-bacteria control group was employed in the biodegradation experiment of 90 days and also showed a trend of weak weight loss (~ 3% to 5%, Figure 2c). The data of weight loss of PVC film in the manuscript was calculated on the basis that had excluded the weight loss of the control group (no bacteria), and

provided in Figure 2c. The calculation formula was described in Supplementary Method S3.

9. Line 343: Since the protein was not recombinantly expressed and characterized, it should be stated that this is a hypothesis. We do not know if this protein degrades PVC.

Responses: Thanks for your important comment. We supplemented the recombinant expression experiments and verified the degradation activity of catalase-peroxidase on PVC film. Please refer to the functional verification results in Line 335-351 and Line 658-676 of the revised manuscript.

10. Line 347: The authors describe that laccase, monooxygenase, and dioxygenases process the degradation products, but then go on to state that these enzymes were not detected in the proteome on line 353, which indicates that the former claim is not supported. Were these genes upregulated in the transcriptome?

Responses: Thanks for the comments. First of all, we wish to clarify that our ascription of laccase, and monooxygenase to PVC degradation products was mainly based on the presence of their encoding genes in the strain genome and prior literature report on their contribution to PE biodegradation. Due to the lack of real-time detection throughout our experiments, these enzymes were likely to be undetectable in the proteome. On the other hand, the degradation of PVC was a dynamic process, hence it was possible that these enzymes had been largely consumed during sampling when their contents were actually below the detection limit. Therefore, this situation is not necessarily contradictory.

Regarding transcriptome analysis, the whole degradation process was only sampled by the end of 10 days without time-series sampling or real-time detection. Therefore, the specific time points when the genes of these enzymes (i.e., laccase, monooxygenase) were up-regulated by the strain were unfortunately not captured. So we can only make a reasonable functional speculation and prediction based on the existing literature survey and cognitive level. In the revised manuscript, we have explained on the above results of the proteome and transcriptome in the Line 379-Line381.

Line 379-Line381.:It is likely that these enzymes are promptly consumed during PVC-

dependent growth of the strain, and/or their expression activities at the single sampling points of our study happened to fall below the detection limit.

11. Line 366: Is there a chance that the larvae could be using the leftover soybean oil in PVC for survival?

Responses: Thanks for the important question. Theoretically, the chance in larva survival with the remaining soybean oil cannot be completely eliminated considered. However, based on our first-hand results, strain EMBL-1 isolated could not utilize soybean oil as the sole carbon and energy source. As you known, the intestinal microbes of larva were diverse, and we cannot completely rule out that some of them might potentially utilize residual soybean oil (if any) in the pre-cleaned PVC film (see Method S2 for the pre-cleaning procedure) for growth and provide energy for the larva, which will also be what we need to examine in a follow-up study. Thank you again for raising this specific but critical point.

12. Generally, the discussion part is inadequate and repeats the results part. Again, the authors mention mRNA, but I did not find a report of transcriptome results or the consolidation of these with the proteome results.

Responses: Thanks for your comment. In the original manuscript, we did have a brief report of transcriptome results in Line316- Line 325. In this revised manuscript, we have followed your guidance to revise and strength the discussion on the transcriptome results and the proteome results (Line 413-422 and Line 450-456).

Line 413-422: These metabolic activities were also manifested by some gene products related to cell growth and death as shown by proteomic (d) and/or transcriptomic (g) analysis (Figure 4). In particular, comparative transcriptomics showed significant upregulation of enzyme-coding genes involved in xenobiotic biodegradation and metabolism (e.g., MBL fold metallohydrolase and alkyl hydroperoxide reductase). Meanwhile, considering the fact that the biodegradation of substances was inseparable from their transportation and catabolism, the significant upregulation of multiple transporter genes (Figure 4g) revealed that the molecular transport ability was closely linked with extracellular PVC polymerization, further conversion,

and eventual intracellular utilization (Figure 5).

Line 450-456: Last but not least, both our proteomic (d) and transcriptomic (f & g) data have witnessed significant expression activities of MFS and ABC family transporters during PVC degradation by strain EBML-1. These transporters play a key role in the uptake of degradation products, consistent with the report by Gravouil et al. (2017) that proteins of the major facilitator superfamily (MFS) or a vector containing an ATP binding cassette (ABC) can integrate low-molecular-weight PE oligomers into cells to achieve PE degradation⁵⁸.

The conclusion part is also repetitive. Especially lines 412-422 are repetitions of the discussion part.

Responses: In the updated conclusion part, we have replaced the content of Line 412- Line 422 with highly condensed contents (Line 461- Line 464) to avoid repetitions of the discussion part as your suggested.

Line 461- Line 464: In this study, we discovered *S. frugiperda* larva feeding on PVC, isolated PVC-degrading *Klebsiella* sp. strain EMBL-1 from their intestinal microbiota, and proposed a hypothetical PVC biodegradation pathway via dechlorination, depolymerization, oxidation and further degradation and mineralization.

13. Figure 3: Panel b is not very informative for the purposes of this paper, and quite hard to distinguish the colors. Removing this panel and extending tree with more species and meta information (where they have been isolated etc) would make the figure more informative.

Responses: Thanks for your helpful comments. Following your suggestions, we have deleted panel b and extended the tree in panel a with more species and meta information (including the places and sample types where they have been isolated) to make Figure 3 more informative.

14. Figure 4: The proteomic and transcriptomic data should be presented comparatively and consolidated (e.g comparison of fold changes between transcriptome and proteome for genes of interest). At the moment it is hard to draw conclusions.

Responses: Thanks for your specific comment. This comment is highly relevant to our

responses to your specific comment 12. We acknowledged that the integration of the transcriptomic and proteomic data still has its rooms for further improvement. In a strict sense, some genes of interest could not be directly comparable between their transcriptomic and proteomic profiles due to the difference in sampling time. Moreover, transcriptome and proteome detection assays usually do not ideally achieve quantitative results, so it is difficult to directly count the fold changes of interest genes in the transcriptome and proteome or draw a solid conclusion on their comparative profiles. Nonetheless, we have tried our best to connect and speculate the useful information in both datasets as much as possible to provide readers some informative reference. We have also followed your helpful guidance in the specific comment 12 to strength the relevant discussion (Line 413-422 and Line 450-456).

What is the difference between the heatmap and the bar chart in terms of what they depict?

Responses: The heatmap shows the relative abundance of 39 proteins in the proteome, while bar chart depicts the values of fold change from the extracellular to intracellular samples. We have improved description on the difference in the updated caption of Figure 4.

15. Figures S3d & S6f, and Figure 4e: It seems like the initial O.D. of the cultures is very high, they were already inoculated at 0.2-0.3. Why was this low dilution factor used? This would lead to the carry over of a high number of late-growth-stage bacteria.

Responses: Thanks for your comments. The initial OD of the liquid culture of the strains was somewhat high (ranging from 0.15 to 0.23) throughout the experiments in our study. We consider this low dilution factor as acceptable because we observed the OD value can be increased to 0.6 or higher in the early stage with the growth of strain EMBL-1 and we replaced the fresh medium regularly, so we think this initial inoculum was normal and could meet with our experimental needs.

Also, for the experiment where the growth in soybean oil was assessed, the starting O.D. of the experiment and the control are different.

Responses: Our deep appreciation on the careful check. For the experiment to evaluate whether EMBL-1 can grow with soybean oil as the only carbon source, the initial inoculation amount was kept the same in the control and experiment groups. The reason for the high OD value in

the treatment group was that soybean oil was insoluble and suspended in the solution after shaking. As you can see, the OD values of the treatment group were always higher than that of the control group by almost the same values. In the revised manuscript, we also supplemented the results of OD changes over time for the MSM+SO group (Figure S3d) to make it easier to understand the results by readers.

Reviewers' Comments:

Reviewer #1:

Remarks to the Author:

Zhe et al. have thoroughly revised the manuscript taking into account the comments by the reviewers.

Overall, I am satisfied with the changes made and their reply to my comments.

Nevertheless, the following minor revisions are still required (beside careful polishing of the English language at certain locations in the manuscript):

p. 14, lines 336-339 are identical (!) to lines 353-356 on the same page. I suggest to delete the text in lines 336-339.

p.17, line 425: ... since the discovery of PVC in 1835...: I am sure that PVC was not 'discovered', but synthesized for the first time in 1835. It is not a natural product that can be 'discovered'

Figure 2g: please fix the chemical structures. The C_xH_n given on the left side of the molecules must be adjusted as not the hydrogen is bound to the alkane/ester chain, but the carbon. Thus "C₈H₁₇-" must read "H₁₇C₈-". The trick in ChemDraw is to type this in the text field at the bond, then choose "Text", "Flush right" from the pull-down menu.

Dehalogenases: I appreciate that they looked into the occurrence of dehalogenases and also mention chloroperoxidases. However, I am rather convinced that not a chloroperoxidase is involved in the reaction shown in Figure 5. A chloroperoxidase does not catalyze the "dehalogenation" proposed there, it is an oxidative enzyme, which then must lead to other products. In turn certain dehalogenases can directly convert a halogenated compound (like the chlorinated compound shown here) into the corresponding alcohol (with water). Thus, Figure 5 needs to be corrected and the text adjusted.

Figure 5, right upper panel, similar comment as above, do not write "C₂₄-C₅", but also add the hydrogens to the formula.

Reference list: sometimes paper titles are given in capital letters, mostly not, this must be adjusted.

Table 1: *E. coli* must be given in italic font

Ref. 4: name of microbe must be in italic font

Reviewer #2:

Remarks to the Author:

The reviewer #2 recognizes significant improvements in the revised manuscript. In particular, biochemical demonstration of a catalase-peroxidase activity towards PVC is highly valued.

Specific comments:

(1) Line 39, What types of "promiscuous" activities here?

(2) Line 246-, I am still missing the meaning of this growth experiment using SO as a carbon source. Are the authors afraid of contamination of SO to PVC?

(3) S8a, Missing in the text.

(4) Line 342-343, Missing figures to show these results.

Reviewer #3:

Remarks to the Author:

The current version of the manuscript is much improved compared to the initial one. It is still a bit concerning that the enzymes that are proposed to degrade the PVC (such as the laccases) were not detected in the proteome, in our experience the plastic-degradation genes are always among the most abundant in the proteome. Therefore, it is not quite clear at the moment if these are responsible for degradation. Also, an enzyme was purified and its PVC degradation was assessed, but the authors have not detected any degradation products using GC-MS or LC-MS, which would have been more solid evidence for the activity of the enzyme.

Responses to the Reviewers

Dear Reviewers,

Thank you for your useful comments on our paper entitled “Polyvinyl Chloride Degradation by Intestinal *Klebsiella* of Pest larvae” (Manuscript ID: NCOMMS-22-07264B). We have revised the manuscript according to your precious comments. Please see the revision with tracked changes marked in red in the re-submitted version and the point-by-point responses to the comments below.

Reviewer #1:

Zhe et al. have thoroughly revised the manuscript taking into account the comments by the reviewers.

Overall, I am satisfied with the changes made and their reply to my comments.

Nevertheless, the following minor revisions are still required (beside careful polishing of the English language at certain locations in the manuscript):

p. 14, lines 336-339 are identical (!) to lines 353-356 on the same page. I suggest to delete the text in lines 336-339.

Response: We sincerely appreciate you for your time in reviewing the manuscript and giving your valuable comments. We have followed your suggestion to delete the text in lines 336-339.

p.17, line 425: ... since the discovery of PVC in 1835...: I am sure that PVC was not ‘discovered’, but synthesized for the first time in 1835. It is not a natural product that can be ‘discovered’.

Responses: Thanks for raising the important comments. We have revised ‘discovered’ as ‘first synthesis’ according to your suggestion (Line 427).

Figure 2g: please fix the chemical structures. The C_xH_n given on the left side of the molecules must be adjusted as not the hydrogen is bound to the alkane/ester chain, but the carbon. Thus

“C8H17-“ must read “H17C8-“. The trick in ChemDraw is to type this in the text field at the bond, then choose “Text”, “Flush right” from the pull-down menu.

Responses: Thanks for the careful check and insightful comments. Following your suggestion, we have fixed the chemical structures in Figure 2g. We sincerely appreciate your shared experience with the proficient use of ChemDraw.

Dehalogenases: I appreciate that they looked into the occurrence of dehalogenases and also mention chloroperoxidases. However, I am rather convinced that not a chloroperoxidase is involved in the reaction shown in Figure 5. A chloroperoxidase does not catalyze the “dehalogenation” proposed there, it is an oxidative enzyme, which then must lead to other products. In turn certain dehalogenases can directly convert a halogenated compound (like the chlorinated compound shown here) into the corresponding alcohol (with water). Thus, Figure 5 needs to be corrected and the text adjusted.

Responses: We appreciate your comment and revise as suggested. Your suggestion made us know more about dehalogenase. We have revised the text about ‘chloroperoxidase’ as ‘dehalogenases’ (Line 35 and Line 444-445) and corrected accordingly in the Figure 5.

Figure 5, right upper panel, similar comment as above, do not write “C24-C5”, but also add the hydrogens to the formula.

Responses: Thank you for your comments. We have revised the formula as your suggestion.

Reference list: sometimes paper titles are given in capital letters, mostly not, this must be adjusted.

Responses: Thanks and revised as suggested.

Table 1: *E. coli* must be given in italic font.

Responses: Thanks and revised as suggested.

Ref. 4: name of microbe must be in italic font.

Responses: Thanks and revised as suggested.

Reviewer #2 (Remarks to the Author):The reviewer #2 recognizes significant improvements in the revised manuscript. In particular, biochemical demonstration of a catalase-peroxidase activity towards PVC is highly valued.

Responses: We sincerely appreciate you for your time in reviewing the manuscript and giving your valuable comments for its further improvement.

Specific comments:

(1) Line 39, What types of “promiscuous” activities here?

Responses: We appreciate your comment and question. The function of catalase-peroxidase itself is to catalyze redox reaction, and the original substrate of the reaction is H₂O₂. In this study, our experiment shows depolymerization activity of this enzyme to PVC, so we think this property belongs to the ‘catalytic promiscuous’ activities of the enzyme.

(2) Line 246-, I am still missing the meaning of this growth experiment using SO as a carbon source. Are the authors afraid of contamination of SO to PVC?

Responses: Thanks for raising this important question. Frankly, we are not afraid of the pollution of SO on PVC, but just aim to eliminate the potential interference of residual SO (if any on the precleaned film) on the growth of EMBL-1 strain, so as to better support that EMBL-1 can grow by using PVC alone as the only carbon source.

(3) S8a, Missing in the text.

Responses: Thanks for raising this point. The text existed in Line 287.

(4) Line 342-343, Missing figures to show these results.

Responses: We appreciate your comment and question. This enzyme activity is directly detected and calculated by using the kit according to the manufacturer’s instructions. We have added the specific method used for determining the enzyme activity in the supplementary information, hoping to help you and readers better understand the contents (Supplementary Information Line 250-256).

Reviewer #3 (Remarks to the Author):

The current version of the manuscript is much improved compared to the initial one. It is still a bit concerning that the enzymes that are proposed to degrade the PVC (such as the laccases) were not detected in the proteome, in our experience the plastic-degradation genes are always among the most abundant in the proteome. Therefore, it is not quite clear at the moment if these are responsible for degradation. Also, an enzyme was purified and its PVC degradation was assessed, but the authors have not detected any degradation products using GC-MS or LC-MS, which would have been more solid evidence for the activity of the enzyme.

Responses: We sincerely appreciate you for your time in reviewing our manuscript and offering your valuable comments and suggestions. We have thoroughly considered your concerns or comments and addressed every one of them as best as we can. Following your suggestion, we conducted new experiments and used GC-MS to detect the possible enzymatic degradation products of PVC, and identified 4 kinds of potential degradation products in the reaction solution of PVC and catalase-peroxidase. The content of the corresponding part has been added to the Results (Line348-353), Methods (Line 678- 680) and Supplementary Information (SI, Line 282- 295, Figure S10), hoping to make the degradation of PVC more conclusive. As for you're a little concern on the absence of laccase in the proteome, we had also discussed that this may be related to the timing we collected the sample in Line 378-383 of the revised manuscript.

Reviewers' Comments:

Reviewer #3:

Remarks to the Author:

[No further comments]